# Follow-Up Study Confirms the Presence of Gastric Cancer DNA Methylation Hallmarks in High-Risk Precursor Lesions

**DOI:** 10.3390/cancers13112760

**Published:** 2021-06-02

**Authors:** Antonio Gómez, Miguel L. Pato, Luis Bujanda, Núria Sala, Osmel Companioni, Ángel Cosme, Martina Tufano, David J. Hanly, Nadia García, José Miguel Sanz-Anquela, Javier P. Gisbert, Consuelo López, José Ignacio Elizalde, Miriam Cuatrecasas, Victoria Andreu, María José Paules, María Dolores Martín-Arranz, Luis Ortega, Elvira Poves, Jesús Barrio, María Ángeles Torres, Guillermo Muñoz, Ángel Ferrández, María José Ramírez-Lázaro, Sergio Lario, Carlos A González, Manel Esteller, María Berdasco

**Affiliations:** 1Cancer Epigenetics Group, Cancer Epigenetics and Biology Program (PEBC), Bellvitge Institute for Biomedical Research (IDIBELL), 08908 Barcelona, Spain; antonio.gomez@vhir.org (A.G.); mlopez@carrerasresearch.org (M.L.P.); martina.tufano@unina.it (M.T.); dhanly@carrerasresearch.org (D.J.H.); 2Epigenetic Therapies Group, Experimental and Clinical Hematology Program (PHEC), Josep Carreras Leukaemia Research Institute, 08916 Barcelona, Spain; 3Department of Gastroenterology, Hospital Donostia/Instituto Biodonostia, Universidad del País Vasco (UPV/EHU), Centro de Investigación Biomédica en Red de Enfermedades Hepáticas y Digestivas CIBEREHD, 20014 San Sebastián, Spain; bujanda@chdo.osakidetza.net (L.B.); angel.cosmejimenez@osakidetza.net (Á.C.); 4Unit of Nutrition, Environment and Cancer, Institut Català d’Oncología, 08908 Barcelona, Spain; nsala@iconcologia.net (N.S.); ocompanioni@iconcologia.net (O.C.); ngarcia@idibell.onmicrosoft.com (N.G.); cagonzalez@iconcologia.net (C.A.G.); 5Translational Research Laboratory, Catalan Institute of Oncology (ICO)-IDIBELL, 08908 Barcelona, Spain; 6Department of Pathology, Hospital Universitario Príncipe de Asturias, 28805 Alcalá de Henares, Spain; josemiguel.sanz@salud.madrid.org; 7Department of Gastroenterology, Hospital Universitario de La Princesa, Instituto de Investigación Sanitaria Princesa (IIS-IP), Universidad Autónoma de Madrid, and Centro de Investigación Biomédica en Red de Enfermedades Hepáticas y Digestivas (CIBEREHD), 28006 Madrid, Spain; javier.p.gisbert@gmail.com (J.P.G.); consuelo.lopezel@salud.madrid.org (C.L.); 8Department of Gastroenterology, Hospital Clínic de Barcelona, IDIBAPS and CIBEREHD, 08036 Barcelona, Spain; elizalde@clinic.cat; 9Department of Pathology, Hospital Clínic de Barcelona, IDIBAPS and CIBEREHD, 08036 Barcelona, Spain; MCUATREC@clinic.ub.es; 10Department of Gastroenterology, Hospital de Viladecans, 08840 Barcelona, Spain; vandreu.hv@gencat.cat; 11Department of Pathology, Hospital Universitari de Bellvitge, 08907 L’Hospitalet de Llobregat, Spain; mjpaules@bellvitgehospital.cat; 12Department of Gastroenterology, Hospital Universitario La Paz, Instituto de Investigación Sanitaria La Paz (IdiPaz), 28046 Madrid, Spain; mmartinarranz@salud.madrid.org; 13Department of Gastroenterology, Hospital Clínico San Carlos, 28040 Madrid, Spain; luis.ortega@salud.madrid.org; 14Department of Gastroenterology, Hospital Universitario Príncipe de Asturias, 28805 Alcalá de Henares, Spain; epoves.hupa@salud.madrid.org; 15Department of Gastroenterology, Hospital Universitario Río Hortega, 47012 Valladolid, Spain; jbarrio@saludcastillayleon.es; 16Department of Pathology, Hospital Universitario Río Hortega, 47012 Valladolid, Spain; mtorresni@saludcastillayleon.es; 17Department of Gastroenterology, Hospital Clínico Universitario Lozano Blesa, 50009 Zaragoza, Spain; gmunnozgon@salud.aragon.es (G.M.); angel.ferrandez@telefonica.net (Á.F.); 18Department of Medicine, Digestive Diseases Service, Institut Universitari Parc Taulí, 08201 Sabadell, Spain; MRamirezL@tauli.cat (M.J.R.-L.); slario@tauli.cat (S.L.); 19Cancer Epigenetics Group, Cancer and Leukemia Epigenetics and Biology Program (PEBCL), Josep Carreras Leukaemia Research Institute (IJC), 08916 Barcelona, Spain; mesteller@carrerasresearch.org; 20Centro de Investigación Biomédica en Red Cáncer (CIBERONC), 28029 Madrid, Spain; 21Institució Catalana de Recerca i Estudis Avançats (ICREA), 08010 Barcelona, Spain; 22Physiological Sciences Department, School of Medicine and Health Sciences, University of Barcelona, 08036 Barcelona, Spain

**Keywords:** precursor lesions, intestinal type of gastric cancer, CpG methylation, *Helicobacter pylori*, cancer risk prediction

## Abstract

**Simple Summary:**

Intestinal metaplasia confers an increased risk of progression to gastric cancer. However, some intestinal metaplasia patients do not develop cancer. The development of robust molecular biomarkers to stratify patients with advanced gastric precursor lesions at risk of cancer progression will contribute to guiding programs for prevention. Starting from a genome-wide methylation study, we have simplified the detection method regarding candidate-methylation tests to improve their applicability in the clinical environment. We identified CpG methylation at the *ZNF793* and *RPRM* promoters as a common event in intestinal metaplasia and intestinal forms of gastric cancer. Furthermore, we also showed that *Helicobacter pylori* infection influences DNA methylation in early precursor lesions but not in intestinal metaplasia, suggesting that therapeutic strategies to prevent epigenome reprogramming toward a cancer signature need to be adopted early in the precursor cascade.

**Abstract:**

To adopt prevention strategies in gastric cancer, it is imperative to develop robust biomarkers with acceptable costs and feasibility in clinical practice to stratified populations according to risk scores. With this aim, we applied an unbiased genome-wide CpG methylation approach to a discovery cohort composed of gastric cancer (*n* = 24), and non-malignant precursor lesions (*n* = 64). Then, candidate-methylation approaches were performed in a validation cohort of precursor lesions obtained from an observational longitudinal study (*n* = 264), with a 12-year follow-up to identify repression or progression cases. *H. pylori* stratification and histology were considered to determine their influence on the methylation dynamics. As a result, we ascertained that intestinal metaplasia partially recapitulates patterns of aberrant methylation of intestinal type of gastric cancer, independently of the *H. pylori* status. Two epigenetically regulated genes in cancer, *RPRM* and *ZNF793*, consistently showed increased methylation in intestinal metaplasia with respect to earlier precursor lesions. In summary, our result supports the need to investigate the practical utilities of the quantification of DNA methylation in candidate genes as a marker for disease progression. In addition, the *H. pylori*-dependent methylation in intestinal metaplasia suggests that pharmacological treatments aimed at *H. pylori* eradication in the late stages of precursor lesions do not prevent epigenome reprogramming toward a cancer signature.

## 1. Introduction

Gastric cancer remains the third leading cause worldwide of cancer death in men and the fifth leading cause in women [1]. Surgical resection of the primary tumor offers the only chance of cure in the early stage of the disease; however, surgery is not an option in advanced gastric tumors that show a poor prognosis [2]. In consequence, the main strategies in gastric cancer management are first, prevention, i.e., the identification of individuals at the highest risk of cancer initiation, and second, early diagnosis [3].

According to Lauren’s classification [4] there are two main histological subtypes of gastric cancer: intestinal subtype, gland-forming adenocarcinoma, and diffuse subtype, with a high presence of non-cohesive infiltrating cells. The accepted mechanism of progression to the intestinal type of gastric cancer is a multistep injury of the gastric mucosa, the so-called “gastric precancerous cascade” [5]. The first alteration in the gastric mucosa is characterized by an active chronic inflammation that can progress to multifocal chronic atrophic gastritis (CAG). CAG is considered the first step in the precancerous cascade that can sequentially progress to intestinal metaplasia (IM; complete or incomplete subtype), dysplasia (low- or high-grade) and, finally, invasive gastric carcinoma. One of the main risk factors that drive the precancerous cascade by initiating the first lesion or non-atrophic gastritis (NAG) [6] is *Helicobacter pylori* (*H. pylori*) infection, and pharmacological eradication of the bacterium in infected patients is one of the most effective prevention strategies in gastric cancer. It is well known that *H. pylori* infection causes changes in the CpG methylation levels of specific genes in the gastric mucosa [7], but it is still to be determined how stable these epigenetic alterations are during progression in the precancerous cascade. This stability would influence the effectiveness of the *H. pylori* eradication strategies.

Since CpG methylation is altered both in precursor lesions and gastric cancer [8], we hypothesize that aberrant methylation in cancer can be partially initiated in precursor lesions. To address this question, we analyzed epigenomic (DNA methylation), pathogen (*H. pylori* infection) and histological data from early to advanced precursor lesions as well as gastric cancer. The results will help to stratify those patients with advanced precursor lesions and at risk of cancer progression, and to guide prevention strategies for their disease.

## 2. Materials and Methods

### 2.1. Patients and Samples

All human samples included in the study were obtained after their respective institutional review board and ethical approval (Ethics Committee at Bellvitge Hospital Ref. PR073/10).

Primary tumors from gastric adenocarcinomas were obtained from the Basque Biobank for Research-OEHUN (Ref. CBVI239). Two sample pieces were extracted from each patient: one corresponding to the tumor and the other from adjacent non-tumor gastric mucosae from the same patient. A pathologist performed the histological diagnosis. For each block, slides were prepared and analyzed using the hematoxylin and eosin, alcian blue (pH 2.5)-periodic acid Schiff (AB-PAS), and modified Giemsa stains. The clinical parameters of the 24 patients (13 intestinal and 11 diffuse subtypes) included in the study are summarized in Appendix A.

Fresh samples from precursor lesions were obtained from biopsies taken from patients at the Gastroenterology Service at Hospital Clinic and Viladecans Hospital (Barcelona). Histological diagnoses were performed according to established guidelines [9] that distinguish the following categories: normal mucosae (NM; *n* = 10), non-atrophic gastritis (NAG; *n* = 10), multifocal chronic atrophic gastritis (CAG; *n* = 13), complete or predominant complete intestinal metaplasia (CIM; *n* = 12), and incomplete or predominant incomplete intestinal metaplasia (IIM; *n* = 19). The clinical parameters of the precursor lesion samples included in the study are summarized in Appendix A.

Paraffin-embedded samples from precursor lesions (*n* = 264) were obtained from an observational longitudinal study [9,10]. During 2012 and 2013, all patients with preliminary histological diagnoses of CAG, IM or dysplasia between 1995 and 2004 were identified from the Pathology Department’s files at nine Spanish National Health Services Hospitals. Inclusion criteria were: 25–69 years of age, absence of peptic ulcer, Barrett’s esophagus, gastric cancer, other cancer or gastric resection, and available clinical and demographic information. The follow-up of these patients during a 12-year period was annotated to identify progression or regression events [9,10]. In brief, five specimens from the same patient were obtained according to the Sidney recommendations (one from the incisura angularis, two from the antrum, and two from the corpus of the stomach). Diagnoses were made with one slide stained with hematoxylin and eosin, and biopsy fragments were classified according to the most advanced lesion in the Correa cascade. Only the sample from antrum was selected for the methylation array study to avoid anatomic region-specific methylation. The same pathologist from each hospital reviewed the initial and final biopsies, following the standard protocol for the diagnosis and classification of cases. The lesions were scored as follows: 1 = normal, 2 = non-atrophic gastritis (NAG), 3 = non-metaplastic multifocal atrophic gastritis (CAG), 4 = complete intestinal metaplasia including predominant complete (CIM), 5 = incomplete intestinal metaplasia including predominant incomplete (IIM), 6 = dysplasia, and 7 = gastric cancer (GC). We considered that the lesions had progressed or regressed if they had, respectively, advanced or regressed at least one point in the overall score (1 to 7). The lesions were stable if they maintained the same score.

### 2.2. Cell Line Culture, Demethylation Treatments and Gene Expression Analysis

The human gastric cancer cell lines (AGS, GCIY and KATO III) were obtained from the American Type Culture Collection (ATCC) (Rockland, MD, USA). Cell lines were maintained in DMEM supplemented with 10% FBS culture media and treated with 1 μM 5-aza-2′-deoxycytidine ( Merck KGaA, Darmstadt, Germany) for 72 h to achieve demethylation. Total RNA was prepared from all samples using TRIZOL^®^ (Invitrogen, Carlsbad, CA, USA) and further purified using RNeasy columns (Qiagen, GmbH, Germany) according to the manufacturer’s instructions. For quantitative RT-PCR assays, 2 μg of total RNA were converted to cDNA with the ThermoScriptTM RT-PCR System (Invitrogen, CA, USA) using Oligo-dT as primer. PCR amplifications were performed as follows: 0.20 μg of cDNA; 5 pmol of each primer and SYBRGreen PCR Master Mix (Applied Biosystems, Foster City, CA, USA). Three measurements were analyzed using a Prism 7700 Sequence Detection (Applied Biosystems, Foster City, CA, USA) instrument. All qRT-PCR were normalized using GAPDH expression as endogenous control. qRT-PCR primer sequences were: for *ZNF793*, 5′- CCA AGA GTG AGG CTG GTT TC -3′ (sense) and 5′- CCA AGG TCC TGT GGC TCT AA -3′ (antisense); for *RPRM*, 5′- CCC GCC AAG TTC CAA CAG -3′ (sense) and 5′- CTG TTG GCC AGG AAC AGG -3′ (antisense); for GAPDH, 5′-GAAGGTGAAGGTCGGAGTCA-3′ (sense) and 5′-TGGACTCCACGACGTACTCA-3′ (antisense).

### 2.3. Helicobacter Pylori Infection and Genotyping

*H. pylori* status at recruitment was determined by *cagA* and *vacA* genotyping. After checking for the integrity of FFPE DNA by means of ACTB gene amplification, the *ureA* gene and the *vacAs*, *vacAm* and *cagA* virulence factor genes of *H. pylori* were genotyped by PCR amplification in a 7500 Fast Real-Time PCR System (Applied Biosystems, Foster City, CA, USA). Details on *cagA* and *vacA* genotyping are described in [10]. Infection by *H. pylori* was considered positive if at least two PCR were positive among *ureA*, *vacAs*, *vacAm* and *cagA*. When only one or none of the PCR was positive, infection by *H. pylori* was considered negative. Positive *H. pylori* samples were subdivided into high virulence (HV) vacAs1m1-cagA+ (haplotype s1m1-1) and low–intermediate virulence (LIV) including all cagA-vacA haplotypes different from s1m1-1.

### 2.4. DNA Extraction, Bisulfite Modification and INFINIUM 450K Methylation Array

DNAs were extracted using conventional Phenol:Chloroform:Isoamyl Alcohol (Merck KGaA, Darmstadt, Germany). Bisulfite modification of 600 ng genomic DNA was carried out with the EZ DNA Methylation Kit (Zymo Research, Irvine, CA, USA) following the manufacturer’s protocol. Next, 4 μL of bisulfite-converted DNA was used to hybridize on Infinium HumanMethylation450 BeadChip (Illumina, San Diego, CA, USA, following Illumina Infinium HD methylation protocol. A three-step normalization procedure was performed using the lumi [11] package available for Bioconductor [12], under the R statistical environment, consisting of color bias adjustment, background level adjustment and quantile normalization across arrays [13]. The methylation score of each CpG is represented as a β-value. The DNA methylation microarray data are freely available for download from the NCBI Gene Expression Omnibus under accession number GSE127857. 

### 2.5. Hierarchical Cluster Analysis and Definition of CpG Methylation Differences

Before analyzing data, we excluded possible sources of biological and technical biases that could have affected the results (probes located in X/Y chromosomes, SNPs, etc). Additionally, we evaluated the detection probabilities (comparing signal intensities against background noise) for all CpGs and excluded those CpGs with values of *p* > 0.01 in more than one sample. Samples were clustered in an unsupervised manner using the 5000 most variable β-values for CpG methylation, according to standard deviation in the CpG sites located in promoter regions by hierarchical clustering. An agglomeration method of Manhattan distances was used.

For the differential methylation analysis between each condition (see main text), a linear model was built using the minfi library [14] under the R statistical software for all the CpGs. The resulting *p*-values were corrected by multiple testing procedures (FDR). The CpGs selected were those which had an adjusted *p*-value below 0.05 and an absolute methylation differential value over 0.33.

### 2.6. Gene Ontology Analysis

We employed the Bioconductor package GOStats and clusterProfiler [15] under R statistical language to search for overrepresented Gene Ontology biological processes using Fisher’s exact test for obtaining *p*-values for each pathway. In order to control the rate of errors, we corrected with multiple testing using the Benjamini–Hochberg algorithm.

### 2.7. Methylation-Site Promoter (MSP) of Candidate Genes

We determined the CpG island methylation status of the candidate genes in a panel of paraffin-embedded samples (*n* = 258 and *n* = 169 for *ZNF793* and *RPRM*, respectively) by methylation-specific PCR (MSP), using primers specific for either the methylated or modified unmethylated DNA. DNA from normal lymphocytes in vitro, treated with M.SssI methyltransferase, was used as a positive control for methylated alleles. DNA from normal lymphocytes was used as a positive control for unmethylated alleles. *ZNF793* primer sequences for the methylated reaction were 5′- TCG GTT ATT TAG GAT GGG AC -3′ (sense) and 5′- AAC CGT TTC TCA AAC CGT AC-3′ (antisense), and for the unmethylated reaction, primer sequences were 5′- GTA TTG GTT ATT TAG GAT GGG AT -3′ (sense) and 5′- AAC CAT TTC TCA AAC CAT ACC TT -3′ (antisense). *RPRM* primer sequences for the methylated reaction were 5′- GGG TCG TTG TTT GTT TAG C -3′ (sense) and 5′- AAC TCT TCT AAA ACC GTC CG -3′ (antisense), and for the unmethylated reaction, primer sequences were 5′- AAG GGG TTG TTG TTT GTT TAG T -3′ (sense) and 5′- TAA ACT CTT CTA AAA CCA TCC AC -3′ (antisense).

## 3. Results

### 3.1. Sequence-Specific CpG Methylation Reprogramming in the Intestinal Type of Gastric Cancer

We studied the CpG methylation profiles in tumor/adjacent non-tumor paired samples from patients with primary gastric adenocarcinoma classified as an intestinal subtype (*n* = 13) and diffuse subtype (*n* = 11). The distribution of clinical variables in the two groups was analyzed, demonstrating that there was no bias on the two arms of comparison for age (Student’s *t*-test, *p* = 0.2879), for tumor location (Fisher’s exact test, *p* = 0.1789), and for metastasis (Fisher’s exact test, *p*-value = 0.673).

To identify the methylation differences among the normal mucosa and the intestinal or diffuse subtypes at specific sequences, we established two-step data processing (Figure 1A). First, we filtered the samples by applying a threshold of > 0.33 change in average β-values and a false discovery rate (FDR) < 0.01 in an ANOVA test adjusted for multiple testing. As a result, we were able to establish a methylation signature for the diffuse and intestinal subtypes compared to normal mucosa. Specifically, we quantified 48,162 and 34,207 sites as significantly differentially methylated CpG (DMCpG) between the non-tumoral and tumoral tissues for the diffuse and intestinal subtypes, respectively (Figure 1A). As the second level of filtering, we considered a “hypermethylated locus” as a CpG showing a gain of > 25% of methylation in the β-value of the tumor compared with the normal tissue, while a “hypomethylated locus” corresponded with a loss of >25% of the β-methylation value of the tumor compared with the adjacent non-tumoral paired sample. As expected, in both tumor subtypes, hypermethylation is more frequent than a loss of methylation (Figure 1B).

However, there is a more programmed reorganization of the CpG methylation in the intestinal than in the diffuse subtype. First, the intestinal subtype showed a higher widespread hypomethylation of the genome. We identified 358 hypomethylated sequences in the intestinal subtype compared to normal mucosa, while only 169 hypomethylated sequences were identified for the diffuse subtype compared to normal mucosa (Figure 1B). Second, the gain of methylation in the intestinal subtype is more sequence-dependent than in the diffuse subtype. Hypermethylation in the intestinal subtype (*n* = 536) mainly affects the regulatory regions of genes (58% and 22% of DMCpG were in regulatory or body regions, respectively), whereas most of the hyper-DMCpG in the diffuse subtype (*n* = 446) were located in the body of the genes (35% and 43% for regulatory and body, respectively) (Figure 1C). The list of DMCpG showing higher differences between cancer and normal tissues for diffuse or intestinal subtypes are described in Appendix A, respectively. Interestingly, there is a reduced common cancer signature for both subtypes (47 and 38 were hyper- and hypo-DMCpG, respectively). Most of the commonly hyper-DMCpG sequences were located in regulatory regions, and include epigenetic alterations previously described in tumors (Appendix A).

These results support the existence of different molecular mechanisms underlying the intestinal and diffuse subtype of gastric cancer and highlight the need to consider them as different entities in epigenetic-based studies.

### 3.2. Methylation Levels in Gastric Mucosae during the Precursor Lesion Cascade and Its Dependence on Helicobacter Pylori Infection

We performed genome-wide methylation analysis in a discovery cohort of precursor lesions, including: normal mucosae (NM, *n* = 10); non-atrophic gastritis (NAG, *n* = 10), multifocal chronic atrophic gastritis (CAG, *n* = 13) and intestinal metaplasia (IM, *n* = 31) (Figure 2A). The distribution of age and gender in the comparison groups were analyzed, demonstrating that there were no significant differences among precursor lesions in the following comparisons: NM vs. NAG (Student’s *t*-test, *p* = 0.78 for age; Fisher’s exact test, *p* = 0.3007 for gender); NM vs. CAG (Student’s *t*-test, *p* = 0.31 for age; Fisher’s exact test, *p* = 1 for gender) and NM vs. IM (Student’s *t*-test, *p* = 0.26 for age; Fisher’s exact test, *p* = 0.4648 for gender). Unsupervised overall beta values of the precursor lesions do not show significant differences (Figure 2B). Furthermore, differential methylation analysis was conducted using a linear regression model implemented in the limma package for each comparison to be tested: CAG vs. NM, NAG vs. NM and IM vs. NM. After *p*-value false discovery rate (FDR) adjustment, only the IM vs. NM model gave significant results (Appendix A). This result is in accordance with previous reports [7,16], and could be explained by the cellular replacement in intestinal metaplasia regarding previous gastric lesions. Interestingly, hypermethylation is more frequent in regulatory regions, while hypomethylation mainly affects intragenic and open reading frame sequences (Figure 2C), which clearly follows an identical pattern to that described for aberrant CpG methylation in intestinal cancer. Gene Ontology analysis demonstrated a significant enrichment of transcription regulatory functions among genes undergoing a gain of methylation in IM samples (Fisher’s test, Bonferroni correction: transcription regulator activity (GO:0140110), *p*-value = 5.42 × 10^−5^; DNA binding (GO:0003677), *p*-value = 1.00 × 10^−2^; ion binding (GO:0043167), *p*-value= 3.29 × 10^−3^).

Next, we studied the influence of *H. pylori* infection on the CpG methylation events in IM. We stratified the IM cohort in negative, low–intermediate virulent (LIV) strains or highly virulent (HV) strains (Appendix A). The unsupervised clustering of the samples does not allow the classification of uninfected versus infected samples (Figure 2D), nor segregation of infected samples based on the presence of virulent strains (Appendix A). Multivariate regression analysis, considering *H. pylori*-positive versus -negative with sex and age as cofounders, was conducted on IM lesions. After *p*-value FDR adjustment, no significant changes were found. Results demonstrated that *H. pylori* infection cannot explain differences in the global CpG methylome of IM.

We applied the same stratification for CAG samples, and we observed that the unsupervised clustering of samples established two groups, depending on the presence or absence of *H. pylori* (Figure 2E). The list of DMCpG, showing higher differences between infected versus non-infected CAG patients after the *p*-value was adjusted for multiple testing, is shown in Appendix A. Gene ontology analysis demonstrated a significant enrichment of immune-defense and cell-signaling functions among genes undergoing a loss of methylation in infected CAG samples (Fisher’s test, Bonferroni correction: regulation of signaling (GO:0023051), *p*-value = 2.30 × 10^−2^; and defense response (GO:0006952), *p*-value = 1.45 × 10^−2^); whereas the gain of methylation in CAG-infected samples involved cell differentiation and adhesion terms (cell differentiation (GO:0030154), *p*-value = 3.11 × 10^−5^; regulation of biological processes (GO:0050789), *p*-value = 3.27 × 10^−2^).

In summary, we demonstrated that there is a “timing” of genome-wide CpG methylation alterations associated with *H. pylori* infection in the progression of multistep precancerous lesions. *H. pylori* infection should be considered for epigenome analysis in the early stages of a precancerous cascade, but it is not fundamental as the process advances to IM.

### 3.3. Identification of Aberrant Promoter Methylation at RPRM and ZNF793 Genes in the Intestinal Type of Gastric Cancer Previously Established in Intestinal Metaplasia

Although most of the previous research has been focused on genes that gain CpG methylation in gastric cancer with respect to non-malignant stages [17], we decided to look for common CpG methylation events between the IM and intestinal types of gastric cancer, to evaluate their roles as early biomarkers of cancer risk. We focused on hypermethylation events because it is well known that epigenetic silencing of relevant genes by a gain of CpG methylation in regulatory regions is a hallmark of cancer [8], and in addition, we ascertained that a gain of methylation in the intestinal type of gastric cancer was most frequent in regulatory sequences (Figure 1B). We found that only 13 CpG sequences were commonly hypermethylated (≥0.25 delta β-values between disease and normal mucosa) in the IM and intestinal types of gastric cancer, but the list includes probes in the regulatory regions of well-known epigenetically deregulated genes in cancer (Figure 2F), such as: ubiquitin-specific peptidase 44 (*USP44*), a gene frequently hypermethylated in patients with inflammatory bowel disease, who are at high risk of progression to colorectal cancer [18]; ST8 Alpha-N-Acetyl-Neuraminide Alpha-2,8-Sialyltransferase 1 (*ST8SIA1*), a candidate proposed as a biomarker for colorectal screening programs in non-invasive samples [19]; *REPRIMO* (*RPRM*, TP53-dependent G2 arrest mediator homolog gene), a tumor suppressor gene commonly hypermethylated in gastric cancer both in tumor biopsy and blood [20], and zinc finger protein 793 (*ZNF793*), a gene methylated in the preneoplastic Barret’s esophagus disease [21].

Taking this list into consideration, we selected two candidate genes based on the following criteria: first, high methylation similarities in the IM and intestinal types of gastric cancer; second, the previously described roles in gastric tumorigenesis; and third, previous evidence of gene regulation by CpG methylation [20,21]. We selected three CG probes from Illumina methylation arrays upstream of the TSS of *ZNF793* (cg02717801, cg23296010, cg14732998) and three CG probes upstream of the TSS of *RPRM* (cg00341742, cg15400238, cg00143045) (Figure 3A), to further validate the methylation of their regulatory regions. A gain of methylation was observed for *ZNF793* and *RPRM* genes in the intestinal type of gastric cancer compared with their normal paired counterpart (*p* < 0.005) (Figure 3A). We also analyzed the methylation level of *ZNF793* and *RPRM* genes in a set of gastric cancer cell lines (*n* = 31) from which we had in-house methylation data (freely available at GEO: GSE68379) [22]. Appendix A showed methylation values using the three DMCpG probes for *ZNF793* (cg02717801, cg23296010, cg14732998) and *RPRM* (cg00341742, cg15400238, cg00143045), obtained from Infinium 450K methylation arrays. Results confirmed that *ZNF793* and *RPRM* CpG methylation is a frequent event in gastric cancer (Appendix A). To test the functional effect of promoter methylation, we performed treatments with the DNA demethylating agent 5-Azacytidine (AZA) in AGS, GCIY and KATO III gastric cancer cell lines. The reactivation of gene expression in hypermethylated cell lines was possible after 5-AZA treatment (Appendix A).

Next, we investigated the CpG methylation status of *ZNF793* and *RPRM* genes in our discovery cohort of precursor lesions. We observed a significant gain of methylation (two-tailed t-test, *p* < 0.01) in IM, the most advanced precursor lesion, with respect to previous lesions (i.e., CAG or NAG) (Table 1 and Appendix A). Methylation at the *ZNF793* and *RPRM* genes is initiated during IM, and reaches the highest levels in cancer samples (Table 1). We have tested the association of methylation status of CpG sites with potential confounding factors, using the univariate linear regression model (*p* < 0.05 was considered significant). We did not find age, gender or *H. pylori* to be significant confounding variables (*p* = 0.2322 for age, *p* = 0.7741 for gender, *p* = 0.3678 for *H. pylori*). Interestingly, the hypermethylation of *ZNF793* and *RPRM* genes in IM samples was not dependent on the anatomical region (antrum, cardia or body) as confirmed by analyzing an independent cohort of a published dataset (GSE103186) [23] (Figure 3B). In accordance with previous observations, samples from antrum showed a higher methylation level than samples from the cardia or body [23].

We analyzed the methylation status of *RPRM* and *ZNF793* genes in the validation cohort of precursor lesions (*n* = 264) obtained from an observational longitudinal study, with a follow-up of 12 years to identify progression or regression events [9] (Figure 4A). Because the processing of formalin fixed-paraffin-embedded samples could be associated with DNA integrity problems, we were able to obtain high-quality Methylation-Specific PCR (MSP) results for 258 samples for *ZNF793* and 169 samples for *RPRM* (Appendix A). The proportion of methylation described in IM samples was higher than in non-metaplastic samples, both for *ZNF793* (Fisher’s test, *p*-value = 0.009194) and *RPRM* (Fisher’s test, *p*-value = 0.04173) (Figure 4B). When IM histological types are considered, methylation is significant for both genes in IIM (Fisher’s test, *p*-value = 0.006639 for *ZNF793*, *p*-value = 0.004304 for *RPRM*) but not in CIM, where significant differences were found only for *ZNF793* (Fisher’s test, *p*-value = 0.00404 for *ZNF793*, *p*-value = 0.1364 for *RPRM*).

Based on the follow-up during the 12 years of the study, it was considered that the lesions had progressed or regressed if they had, respectively, advanced or regressed at least one point in the precancerous cascade, whilst lesions were considered to be stable if they maintained the same score [9]. It can be observed that methylation in *ZNF793* and *RPRM* was not associated with the progression of any lesion (Figure 4C). This non-significant association was expected, due to the lack of methylation differences at the early stages of the precursor cascade.

From 264 biopsies included in the study, only 9 progressed to gastric cancer after a 12-year follow-up (Figure 4D). This result represents an incidence rate of 2.5 per 1000 persons/year. Accordingly, a systematic review that examined all available evidence of the risk of GC patients with IM has concluded that there is substantial heterogeneity in the incidence, ranging from 0.38 to 17.08 per 1000 persons/year [24]. The low number of cancer patients did not allow any statistical test to check the correlation between methylation and progression. In absolute numbers, we obtained *ZNF793* methylation data for 9 patients and *RPRM* methylation data for 5 patients with cancer progression. Of the patient tumors that progressed from IMM, 50% and 33% were methylated at *ZNF793* (2/4) and *RPRM* (1/3) promoters, respectively.

Our results, by demonstrating the epigenetic-associated silencing of cancer-related genes (*ZNF793* and *RPRM*), provide a molecular mechanism to explain the increased risk of IM progressing to the intestinal type of gastric cancer. The extension of these findings is needed, and future studies in larger populations to increase the number of cases including cancer progression are required.

## 4. Discussion

A strategy leading to the identification of biomarkers at any stage of the multistep precancerous cascade will greatly improve the identification of patients who are at risk of cancer progression. So far, most of the screening programs in patients at moderate–high risk for gastric cancer are mainly based on the status of *H. pylori* infection and the presence and extent of IM (discussed below). In addition to these risk factors, we also recently demonstrated that genetic variability in specific genes (e.g., MUC2, NFKB1 and CD14) is associated with the progression to gastric cancer from precursor lesions [10].

Several examples exist of the use of CpG methylation as a biomarker for patient stratification and disease outcome [8,25,26,27,28]. Regarding two recent examples in diseases affecting the gastrointestinal tract, quantification of the methylation levels in two candidate genes was proposed as a tool for the discrimination of patients with and without Barret’s esophagus disease using non-invasive samples [29], and a panel of methylated genes was also proposed for the discrimination of advanced neoplasia in pancreatic cysts [30]. However, limited studies exist on the alterations of CpG methylation during the progression of the precancerous cascade in gastric cancer, and they refer to the quantification of changes in specific promoters [7,16,17,31]. We have performed a genome-wide CpG methylation analysis of the precancerous cascade, and we have identified an enrichment of regulatory regions in the hyper-DMCpG in IM, which clearly follows an identical pattern to that of aberrant CpG methylation in cancer. We are aware that one of the main difficulties in epigenetic research is to discard tissue specificity and the relative contribution of each cell type [8], an aspect especially relevant for biopsies taken in clinical environments. However, our consistent observation using our retrospective panel of 264 patients showing a difference in the methylation level of *ZNF793* and *RPRM* in IM compared with previous precursor lesions reinforces their potential as candidates for the design of future larger studies. Future single-cell analysis to deconvolute sample heterogeneity should also help to answer this question [32].

Our research design allows us to explore another important question in gastric cancer prevention strategies, that of how determinant *H. pylori* infection is during the progression of the precancerous cascade. Several authors support the “point-of-no-return” hypothesis, in which there is a stage where *H. pylori* eradication does not result in a reversion of the mucosal damage, and, in consequence, does not completely eliminate cancer risk. Although the idea is generally accepted, there is not a consensus on the timing of the “point-of-no-return” [23,33,34]. For some authors, *H. pylori* eradication has a beneficial effect in normal mucosa and NAG, but it has no effect on IM or dysplasia [35]; a clear molecular mechanism demonstrating this hypothesis has not been provided. Our study showed that the alteration of CpG methylation is induced in CAG and maintained during IM. Our observation is in accordance with previous results where no detectable differences in global DNA methylation between high-risk gastric cancer patients with and without active *H. pylori* infection were detected, suggesting that methylation changes are irreversible once it has taken place [36]. It should be noted that other CpG methylation changes associated with cellular trans-differentiation in intestinal metaplasia could occur independently of *H. pylori.* In contrast with our study performed at a loci-specific level, this study was performed by a global DNA quantification method [36]. However, both studies showed the same tendency. In addition, a prospective, randomized, placebo-controlled clinical trial was performed in a high-risk region of China [37]. The study included 1630 patients infected with *H. pylori,* with or without precursor lesions. They found that the incidence of gastric cancer development in the high-risk population after a 7.5-year follow-up was similar between participants receiving *H. pylori* eradication treatment and those receiving the placebo. However, in the subgroup of *H. pylori* patients without precancerous lesions, the eradication of *H. pylori* significantly decreased the development of gastric cancer [37]. These clinical observations are in agreement with our observation that the *H. pylori* presence or absence in earlier lesions (e.g., CAG) influences CpG methylation. Although larger follow-up studies are needed, this evidence opens up the possibility of future studies to address whether the eradication of *H. pylori* by therapeutic strategies before CAG should prevent epigenome reprogramming toward a cancer signature. These strategies could complement current interventions aimed at eradicating *H. pylori* in IM.

Finally, the functional relevance of the two genes showing epigenetic deregulation in IM and gastric cancer should be highlighted. The *ZNF793* gene is especially enticing because methylation of its promoter has been previously proposed as a detection biomarker for Barret’s esophagus, a premalignant condition for esophageal adenocarcinoma [21]. Hypermethylation of *RPRM*, a tumor suppressor gene, has been identified in gastric cancer [38]. It can be detected even in non-invasive samples (serum) and enables discrimination between normal and tumoral patients. This evidence raises hope in the potential of CpG methylation in specific genes as early detection methods.

Although one of the strengths of the paper is its clinical relevance and the access to a screening population set of 264 samples with a follow-up of 12 years, we are aware that additional studies in large cohorts are needed. In our studies, we obtained 9 cases that progressed to cancer, which represents an incidence rate of 2.5 per 1000/person/year. The number of cases that progress to cancer obtained in our study is in accordance with previous estimations in European populations (1.75 cases per person/year) [39]. It should be noted that detection of the CpG methylation of a specific region could be easily performed by simple approaches without complex handling requirements and within an acceptable cost/effectiveness margin. These characteristics facilitate the development of larger studies in clinical environments.

## 5. Conclusions

We present here epigenomic (DNA methylation), pathogen (*H. pylori* infection) and histological data from early to advanced precursor lesions, and from gastric cancer. We hypothesized that specific aberrant methylation observed in gastric cancer can also be detected in early non-tumoral stages corresponding to IM. Importantly, methylation at the specific genes *RPRM* and *ZNF793* in IM is not influenced by *H. pylori* infection, which confers a more universal utility of the epigenetic biomarker and opens up the question of whether therapeutic strategies to eradicate *H. pylori* need to be adopted early in the precursor cascade to prevent cancer epigenetic reprogramming. Herein, we provided molecular clues to further investigate the practical utilities of the quantification of *RPRM* and *ZNF793* DNA methylation level as a marker for gastric cancer risk. Future longitudinal cohort studies will be especially valuable to validate their involvement in gastric cancer progression.

## Figures and Tables

**Figure 1 cancers-13-02760-f001:**
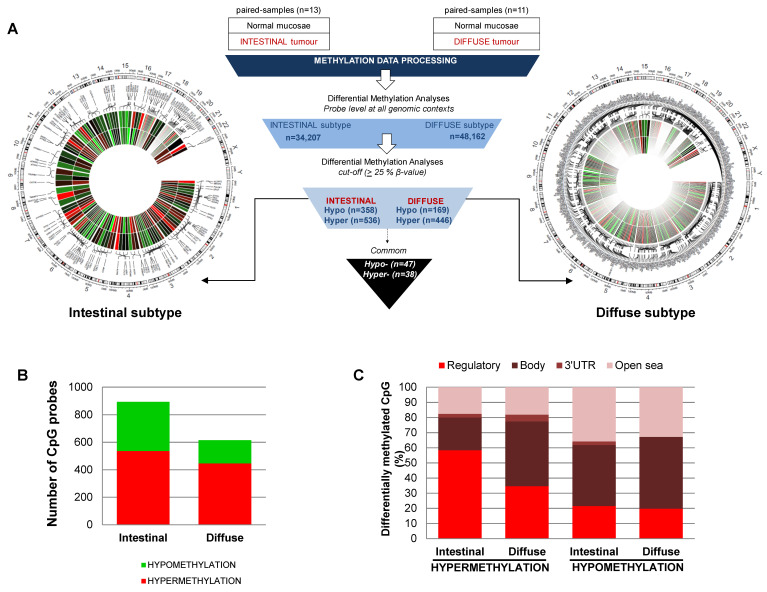
CpG methylation signature of intestinal and diffuse types of gastric cancer. (**A**) Center, workflow of the cancer-related study. Circus graphs for genome-wide DNA methylation levels in diffuse (right) and intestinal (left) gastric tumors. The inner track indicates the β-value of the non-tumoral and the outer circle represents the cancer sample. (**B**) Total number of hypermethylation and hypomethylation events in gastric cancer, relative to adjacent non-tumoral samples. (**C**) Genomic distribution of differentially methylated CpGs in the intestinal and diffuse types of gastric cancer.

**Figure 2 cancers-13-02760-f002:**
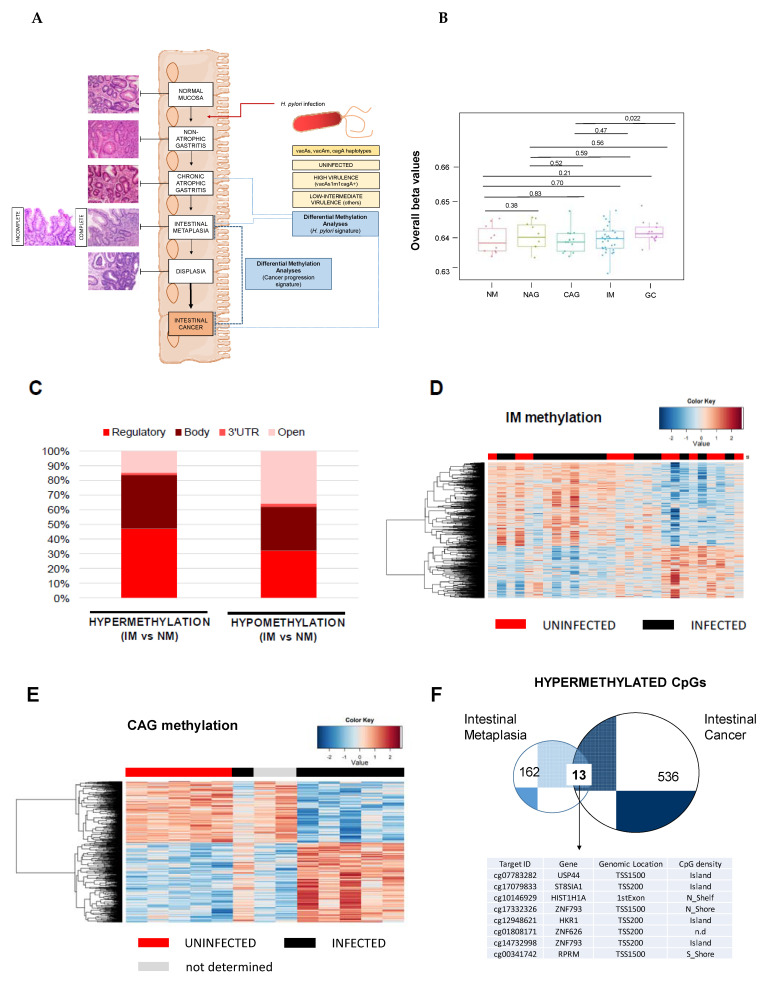
CpG methylation signature of the gastric precancerous cascade. (**A**) Schematic representation of the precursor lesion pathway. Two genome-wide CpG methylation analyses were performed: (1) *H. pylori*-related study to identify the timing of epigenetic aberrations due to bacterial infection, and (2) a study of the hypermethylation events observed in IM that could act as biomarkers of cancer progression. (**B**) Boxplots showing overall β-values for NM, NAG, CAG, IM and GC samples. Overall beta values were calculated as the mean of all CpGs in each sample, then plotted as boxplots for all groups. Each box represents the IQR with horizontal lines representing the median. Whiskers are extended to within 1.5 IQR of the upper and lower quantities. Data points falling outside this range are displayed independently. The *p*-values were calculated using Student’s *t*-test. (**C**) Total number of hypermethylation and hypomethylation events in IM relative to normal tissues. (**D**) Supervised clustering of methylation values in the IM samples stratified by *H. pylori* infection. (**E**) Supervised clustering of methylation values in the CAG samples considering the presence or absence of *H. pylori*. (**F**) Overlapped hypermethylation in IM and intestinal type of gastric cancer. CAG, multifocal chronic atrophic gastritis; GC, gastric cancer; IM, intestinal metaplasia; NAG, non-atrophic gastritis; NM, normal mucosa.

**Figure 3 cancers-13-02760-f003:**
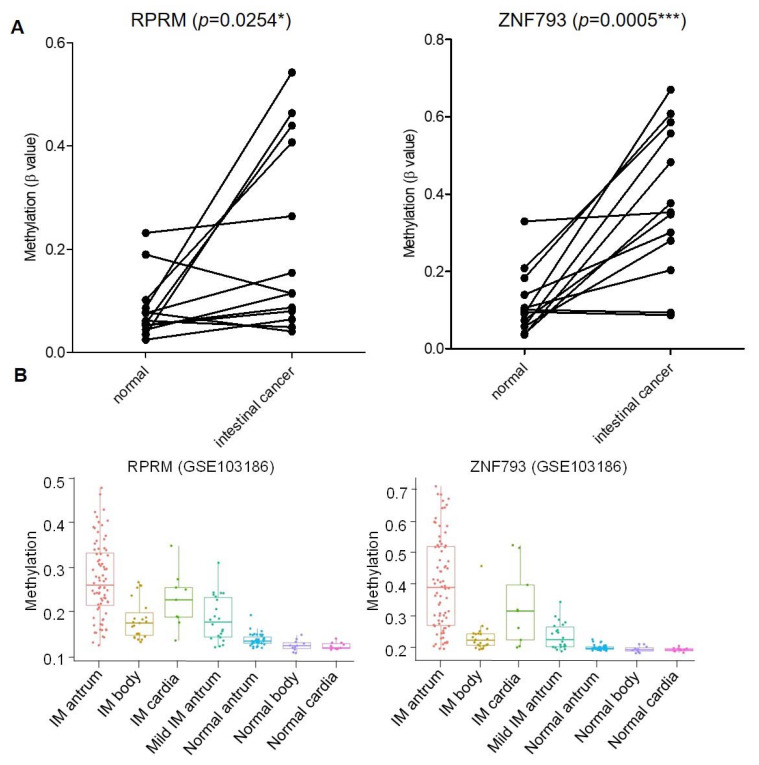
Methylation of *ZNF793* and *RPRM* genes in intestinal type of gastric cancer and precursor lesions. (**A**) Methylation level of *ZNF793* and *RPRM* genes in intestinal type of gastric cancer. Values from 13 adjacent-non-tumoral samples paired with the tumor sample are shown. Methylation values represent the average of β-value obtained for the three CG probes in the methylation array (paired *t*-test, two-tailed, *p* < 0.05, * *p*-value ≤ 0.05, *** *p*-value ≤ 0.001). (**B**) CpG methylation at *RPRM* and *ZNF793* promoter in three anatomic intestinal regions (antrum, cardia and body) of intestinal metaplasia patients and normal mucosae (available data from GSE103186).

**Figure 4 cancers-13-02760-f004:**
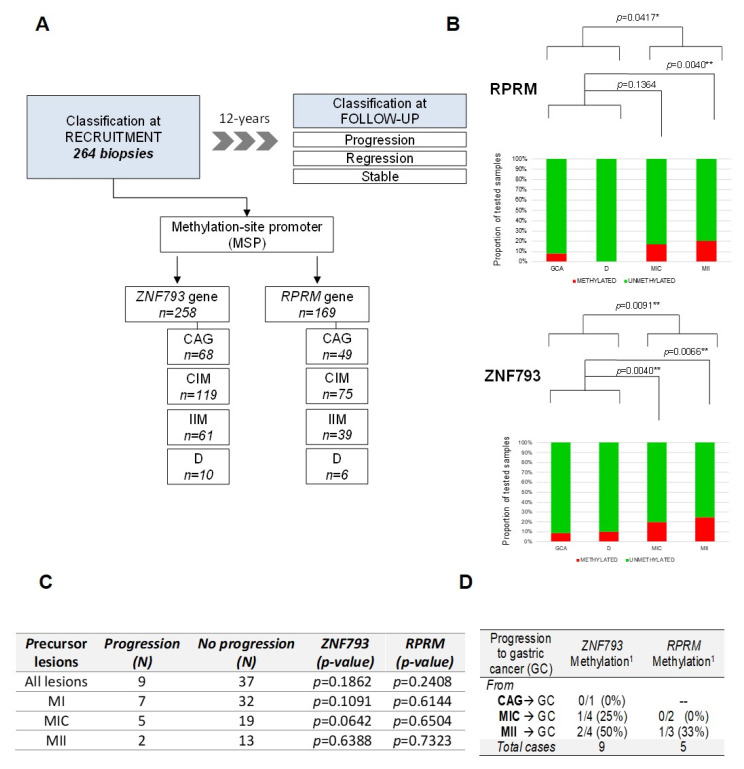
Methylation of ZNF793 and RPRM genes in a 12-year follow-up study of gastric cancer precursor lesions. (**A**) Characteristics of the validation cohort of 264 samples corresponding to precursor lesions obtained from an observational longitudinal study. (**B**) Methylation events for *ZNF793* (*up*) and *RPRM* (*down*) genes in precursor lesions. (**C**) Association analysis between the progression in the cascade of gastric precursor lesions and the methylation of *ZNF793* and *RPRM* genes. The *p*-values are obtained from Fisher’s exact test. (**D**) Number of cases of progression to gastric cancer, after a 12-year follow-up, for gastric precursor lesions. ^1^ Methylation state of *ZNF793* and *RPRM* promoters at the recruitment biopsy are indicated. CAG, multifocal chronic atrophic gastritis; CG, gastric cancer; D, dysplasia; IM, intestinal metaplasia; NAG, non-atrophic gastritis; NM, normal mucosa; * *p*-value ≤ 0.05, ** *p*-value ≤ 0.01.

**Table 1 cancers-13-02760-t001:** Descriptive statistics and results of independent sample *t*-test for significance and comparisons between precursor lesions. * significant at *p* < 0.05, ** significant at *p* < 0.01. CAG, multifocal chronic atrophic gastritis; CIM, complete intestinal metaplasia; GC, gastric cancer; IIM, incomplete intestinal metaplasia; IM, intestinal metaplasia; NAG, non-atrophic gastritis; NM, normal mucosa.

Variable	Group	*n*	AVG	SD	*t*-Test (Two-Tailed)	
					*p*-Value	95% Confidence Level
*RPRM* methylation	NM	10	0.06086331	0.01365057	0.0761	−0.0023, 0.0434
	NAG	10	0.08141519	0.03171394		
	NM	10	0.06086331	0.01365057	0.5917	0.0034 ± 0.0130
	CAG	13	0.06428683	0.0158491		
	NM	10	0.06086331	0.01365057	**0.0092 ****	0.0694 ± 0.0512
	IM	31	0.13023734	0.07911443		
	NM	10	0.06086331	0.01365057	0.0238 *	0.1646 ± 0.1244
	GC	13	0.22550582	0.18777656		
	NAG	10	0.08141519	0.03171394	0.1041	−0.03810, 0.003843
	CAG	13	0.06428683	0.0158491		
	CAG	13	0.06428683	0.0158491	**0.0050 ****	0.020989, 0.11091
	IM	31	0.13023734	0.07911443		
	IM	31	0.13023734	0.07911443	0.0229 *	−0.1766, −0.01389
	GC	13	0.22550582	0.18777656		
	CIM	12	0.10492342	0.02609879	0.1603	−0.09992, 0.01731
	IIM	19	0.14622508	0.0965327		
*ZNF793* methylation	NM	10	0.07191438	0.0149968	0.0175 *	−0.06939, −0.007632
	NAG	10	0.110425403	0.043621525		
	NM	10	0.07191438	0.0149968	0.7792	0.0021 ± 0.0152
	CAG	13	0.07399085	0.01898542		
	NM	10	0.07191438	0.0149968	**0.0041 ****	0.1478 ± 0.0982
	IM	31	0.21970273	0.15204481		
	NM	10	0.07191438	0.0149968	**0.0002 ****	0.3152 ± 0.1311
	GC	13	0.38708098	0.19796844		
	NAG	10	0.110425403	0.043621525	0.0142 *	0.008155, 0.06471
	CAG	13	0.07399085	0.01898542		
	CAG	13	0.07399085	0.01898542	**0.0014 ****	−0.2317, −0.05972
	IM	31	0.21970273	0.15204481		
	IM	31	0.21970273	0.15204481	**0.0049 ****	−0.2811, −0.05364
	GC	13	0.38708098	0.19796844		
	CIM	12	0.19834641	0.1057375	0.5433	−0.1507, 0.8101
	IIM	19	0.23319094	0.17663747		

Bold numbers: Significant *p*-values.

## Data Availability

The dataset supporting the conclusions of this article is available in the Gene Expression Omnibus (GEO) repository under the accession number GSE127857 (private link for the reviewers until the acceptance of the manuscript).

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
