# Peer review of "Follow-Up Study Confirms the Presence of Gastric Cancer DNA Methylation Hallmarks in High-Risk Precursor Lesions"

_cancers, 2021, doi:10.3390/cancers13112760_

Round 1
Reviewer 1 Report
The authors aimed to analyze genome-wide CpG methylation alterations in gastric cancer and precursor lesions and tried to find associations of candidate genes methylation with progression of precursor lesions based on a longitudinal study. Thirteen candidate gene methylations were identified in both IM and intestinal type gastric cancer, but their associations with disease progression were not evaluated except ZNF793 and RPRM. The CpG methylation alterations in gastric cancer also need to be investigated in NAG and CAG in order to find their associations with disease progression. H. pylori infection on methylation should be reanalyzed separately or adjusted in association analysis.
Major comments
- Chi-square test should not be used to compare two groups of consecutive variables.
- The authors identified 48,126 and 34,207 differentially methylated CpG in intestinal type gastric cancer and diffuse type gastric cancer compared with adjacent normal tissues, respectively. After further filtering, 358 Hypomethylation and 536 Hypermethylation were identified for intestinal type gastric cancer, and 169 Hypomethylation and 446 Hypermethylation were identified for intestinal type gastric cancer. And are these methylation differences can be identified in NAG, CAG or IM compared with NM? The common methylation differences in CAG or IM with gastric cancer, especially intestinal type, will be interesting and focused. And detailed data should be shown.
- The authors demonstrated the overall beta values of the precursor lesions did not show significant differences (Figure 2B). How does this reflect high methylation similarities among precursor lesions? Especially in the following paragraph, the authors found CpG methylation increased in IM (Figure 2B).
- The authors used unsupervised clustering to explore the influence of pylori infection in CpG methylation events in IM and demonstrated no differences in the global CpG methylome of IM. Have you tried multivariable analysis to compare infected and uninfected groups in IM to find the differences of CpG methylation, then present by heatmap or tables? The authors demonstrated that they established two groups through unsupervised clustering depending on the presence or absence of H. pylori in CAG. But in Figure 2E, the last sample in the uninfected group and the first two samples in infected group showed obvious different characteristics in methylome. It’s important to mention that there are 31 samples in IM, but only 13 samples in CAG. More samples will make clustering more complicated. So the conclusion of “timing” of genome-wide CpG methylation alterations associated with H. pylori infection in the progression of multistep precancerous lesion is arbitrary.
- I’m confused that I don’t see cell lines data in Figure S3B. And Figure S3B is the same with Figure 3B.
- To investigate the timing of hypermethylation of ZNF793 and RPRM in precursor lesions, all precursor lesions or GC groups should be compared with the same reference (for example, NM group).
- The authors claim that “the proportion of methylation described in IM samples was higher than in non-metaplastic samples both for ZNF793 (Fisher’s test, p-value=0.009194) and RPRM (Fisher’s test, p-value=0.04173) (Figure 4B)”, but these Data are not shown in Figure 4B.
- What’s the method used for association analysis in Figure 4C, and patient numbers of progression and no progression in each group are not shown.
- The authors identified 13 common hypermethylated CpGs between IM and intestinal type gastric cancer. Is there any association of other methylated sites with progression?
Author Response
Reviewer 1
The authors aimed to analyze genome-wide CpG methylation alterations in gastric cancer and precursor lesions and tried to find associations of candidate genes methylation with progression of precursor lesions based on a longitudinal study. Thirteen candidate gene methylations were identified in both IM and intestinal type gastric cancer, but their associations with disease progression were not evaluated except ZNF793 and RPRM. The CpG methylation alterations in gastric cancer also need to be investigated in NAG and CAG in order to find their associations with disease progression. H. pylori infection on methylation should be reanalyzed separately or adjusted in association analysis.
Major comments
Comment R1.1. Chi-square test should not be used to compare two groups of consecutive variables.
Response R1.1. We apologize for the mistake. We have corrected the test for each variable in the new version as follows:
- Page 5: “The distribution of clinical variables in the two groups was analyzed, demonstrating that there was no bias on the two arms of comparison for age (Mann-Whitney U test; p=0.3849), for tumor location (Fisher's exact test, p=0.1789) and for metastasis (Fisher's exact test, p-value= 0.673).”
- Page 6: “The distribution of age and gender in the comparison groups was analyzed, demonstrating that there were no significant differences among precursor lesions in the following comparisons: NM vs NAG (Mann-Whitney U test, p=1 for age; Fisher's exact test, p=0.3007 for gender); NM vs CAG (Mann-Whitney U test, p=0.3513 for age; Fisher's exact test, p=1 for gender) and NM vs IM (Mann-Whitney U test p=0.1464 for age; Fisher's exact test, p=0.4648 for gender)”.
Comment R1.2. The authors identified 48,126 and 34,207 differentially methylated CpG in intestinal type gastric cancer and diffuse type gastric cancer compared with adjacent normal tissues, respectively. After further filtering, 358 Hypomethylation and 536 Hypermethylation were identified for intestinal type gastric cancer, and 169 Hypomethylation and 446 Hypermethylation were identified for intestinal type gastric cancer. And are these methylation differences can be identified in NAG, CAG or IM compared with NM? The common methylation differences in CAG or IM with gastric cancer, especially intestinal type, will be interesting and focused. And detailed data should be shown.
Response R1.2. We thank the comment because the results were not properly explained in the previous version.
Differential methylation analysis was conducted using a linear regression model implemented in the limma package for each comparison to test: CAG vs NM, NAG vs NM and IM vs NM. After p- value adjustment (False Discovery Rate), only the IM vs NM model gave significant results. This evidences that genome-wide methylation does not discriminate “superficial” non-atrophic gastritis or chronic inflammation in gastric mucosae. In consequence, only IM could be compared with cancer methylation.
From a physiological point of view, this could be explained as metaplasia involves a replacement of cell types in the gastric tissue. Epigenetic factors are involved in cell-specification, so methylation differences between metaplasia and inflammation-related phenotypes are justified. In addition, we strongly believe that the inexistence of common DMCpGs in NAG-CAG-IM and cancer is interesting. For example, common methylation in NAG-CAG and intestinal cancer most probably will represent non- cancer specific methylation or methylation associated with the inflammation response. Our increased methylation reprogramming in metaplasia is in agreement with the gradual increase of disease severity, being intestinal metaplasia the precursors with highest risk to cancer progression.
We have modified the text accordingly:
Page 7: “Unsupervised overall beta values of the precursor lesions do not show significant differences (Figure 2B), reflecting the inexistence of a genome-wide reorganization of the methylome initiated at early stages of precursor lesions. Furthermore, differential methylation analysis was conducted using a linear regression model implemented in the limma package for each comparison to test: CAG vs NM, NAG vs NM and IM vs NM. After p- value adjustment (False Discovery Rate), only the IM vs NM model gave significant results (Table S4). This result is in accordance with previous reports [7, 16] and could be explained by the cellular replacement in metaplasia intestinal regarding previous gastric lesions.”
Comment R1.3. The authors demonstrated the overall beta values of the precursor lesions did not show significant differences (Figure 2B). How does this reflect high methylation similarities among precursor lesions? Especially in the following paragraph, the authors found CpG methylation increased in IM (Figure 2B).
Response R1.3. We agree with the reviewer that the text and figures 2B were confusing. We have reordered this information.
First, unsupervised analysis of all CpGs included in the study did not reveal statistical differences in the overall methylation among precursor lesions. This is represented in Figure 2B.
Second, we performed a supervised analysis. As explained in R1.2 no statistical differences were obtained for NAG and CAG but IM revealed DMcPG regarding NM (Figure 2C).
Down panel-Figure 2B from previous version was deleted. It was representing hyper- and hypo-methylation understood as a cut-off: hyper- were CpGs showing β-values higher than 0.75, while hypo- was referred to β-values lower than 0.25. In other words, they were absolute values without comparing with NM. We understand that this information is not adding any value but can lead to misunderstandings. For this reason, we abolished the down panel- Figure 2B in the new version.
In addition, we have included a better definition in Figure Legend 2B:
Page 9: Figure 2. “(B) Boxplots showing overall β-values for normal mucosa (NM), non-atrophic gastritis (NAG), multifocal chronic atrophic gastritis (CAG) and intestinal metaplasia (IM). Overall beta values were calculated as the mean of all CpGs in each sample, then, we plot as boxplots for all groups. Each box represents the IQR with horizontal lines representing the median. Whiskers are extended to within 1.5 IQR of the upper and lower quantities. Data points falling outside this range are displayed independently. p-values were calculated using t-tests.”
Comment R1.4. The authors used unsupervised clustering to explore the influence of pylori infection in CpG methylation events in IM and demonstrated no differences in the global CpG methylome of IM. Have you tried multivariable analysis to compare infected and uninfected groups in IM to find the differences of CpG methylation, then present by heatmap or tables?
The authors demonstrated that they established two groups through unsupervised clustering depending on the presence or absence of H. pylori in CAG. But in Figure 2E, the last sample in the uninfected group and the first two samples in infected group showed obvious different characteristics in methylome. It’s important to mention that there are 31 samples in IM, but only 13 samples in CAG. More samples will make clustering more complicated. So the conclusion of “timing” of genome-wide CpG methylation alterations associated with H. pylori infection in the progression of multistep precancerous lesion is arbitrary.
Answer R1.4. Thanks for the comment.
Following reviewers’ suggestion, we have performed multivariable analysis considering age and sex in infected and uninfected individuals and we did not found significant changes in the methylome of IM samples. We have added the following sentence in results:
Page 7: “Multivariate regression analysis considering H. pylori positive vs negative with sex and age as a cofounders was conducted on IM lesions. After p-value False Discovery Rate (FDR) adjustment, no significant changes were found”.
However, adjusted p-value FDR for uninfected or infected CAG samples showed statistical significant differences in the methylation. The list of DMCpG after analysis are included in Table S5.
We apologize for the mistake in Figure 2E (CAG and H. pylori). The coloured horizontal barr at the top is not correct. As it could be checked in Table S2, we have 13 CAG samples: 5 infected samples, 6 uninfected and 2 samples without information about H.pylori. From the analysis, it seems that these two samples without H.pylori genotyping have methylation profiles close to uninfected patients. However, as we do not have the clinical confirmation we indicate these two samples as “not determined”. Horizontal barr has been corrected accordingly in Figure 2E.
About heatmap, in order to consider the trustability of the clusters, we used Bioconductor packages M3C (John CR, Watson D, Lewis M, Russ D, Goldmann K, Ehrenstein M, Pitzalis C, Barnes M (2018). “M3C: A Monte Carlo reference-based consensus clustering algorithm.” Scientific Reports volume 10, Article number: 1816 (2020). Briefly, M3C is a consensus clustering algorithm that uses a Monte Carlo simulation to eliminate overestimation of K (class assignment) and can reject the null hypothesis K=1 giving a p-value for the best assignment. In our case, K=2 was our best assignment with an associated pval<0.05.
We agree that more studies are needed to gain a better definition of the H. pylori effect. However, our observation is in accordance with previous publications: As example, Leodolter et al (Somatic DNA Hypomethylation in H. pylori-Associated High-Risk Gastritis and Gastric Cancer: Enhanced Somatic Hypomethylation Associates with Advanced Stage Cancer. 2015. PMID 25928808) showed that DNA methylation changes (e.g., global hypomethylation) in H. pylori–related gastritis patients persists after eradication, suggesting that epigenetic changes are irreversible once they have taken place. Indeed, no difference was found between the methylation levels in high-risk gastritis patients with H. pylori present and absent after eradication. Although this result was obtained by a global DNA quantification methods, in contrast with our study performed at loci-specific assay, both showed the same tendency. Similarly, Wong et al (JAMA 2004, Helicobacter pylori eradication to prevent gastric cancer in a high-risk region of China: a randomized controlled trial.PMID 14722144) performed a prospective, randomized, placebo-controlled clinical trial in a high-risk region of China. The study include 1630 patients with H.pylori with or without precursor lesions. They found that the incidence of gastric cancer development at the population level after a 7.5 years follow-up was similar between participants receiving H. pylori eradication treatment and those receiving placebo. However, in the subgroup of H. pylori patients without precancerous lesions, eradication of H.pylori significantly decreased the development of gastric cancer. These clinical observations are in agreement with our observation that CPG methylation in high-risk precursor lesions was not associated with H. pylori presence. In contrast, earlier lesions showed differential methylation between H.pylori positive or negative patients.
We have included the clinical trial study in the discussion:
Page 14: “Our study showed that the alteration of CpG methylation is induced in CAG and maintained during IM. Our observation is in accordance with previous results where no detectable differences in global DNA methylation between high-risk gastric cancer patients with and without active H. pylori infection were detected suggesting that methylation changes are irreversible once it has taken place [36]. In contrast with our study performed at loci-specific level, this study was performed by a global DNA quantification method [36]. However, both studies showed the same tendency. In addition, a prospective, randomized, placebo-controlled clinical trial was performed in a high-risk region of China [37]. The study include 1630 patients infected with H.pylori with or without precursor lesions. They found that the incidence of gastric cancer development at the high-risk population after a 7.5 years follow-up was similar between participants receiving H. pylori eradication treatment and those receiving placebo. However, in the subgroup of H. pylori patients without precancerous lesions, eradication of H.pylori significantly decreased the development of gastric cancer [37]. These clinical observations are in agreement with our observation that H. pylori presence or absence in high-risk precursor lesions does not change DNA methylation. In contrast, earlier lesions showed differential methylation between H.pylori positive or negative patients. Although larger follow-up studies are needed, these evidences opens future studies to address whether eradication of H. pylori by therapeutic strategies before CAG should prevent epigenome reprogramming towards a cancer signature”.
Comment R1.6. I’m confused that I don’t see cell lines data in Figure S3B. And Figure S3B is the same with Figure 3B.
Answer R1.6. We apologize for the mistake. Figure S3 was added wrongly. In the re-submission version we have added the correct document.
Comment R1.7. To investigate the timing of hypermethylation of ZNF793 and RPRM in precursor lesions, all precursor lesions or GC groups should be compared with the same reference (for example, NM group).
Answer R1.7. We agree with the comment. We have now supressed the “timing” for the sentence that has been rephrased as:
Page 10: “Next, we investigated the CpG methylation status of ZNF793 and RPRM genes in our discovery cohort of precursor lesions.”
Our MSP study did not include samples with normal mucosa at the time of recruitment. Paraffin-embedded samples from precursor lesions were obtained from an observational longitudinal study [12,15]. One of the inclusion criteria in the study was preliminary histological diagnoses of CAG, IM or dysplasia. Additional inclusion criteria were 25–69 years of age, absence of peptic ulcer, Barrett’s oesophagus, gastric cancer, other cancer or gastric resection, and available clinical and demographic information. The follow-up of these patients (with CAG, IM or dysplasia at recruitment) during a 12-year period was annotated to identify progression or regression events [12,15]. Our MSP study is qualitative methylation method that aims to correlate unmethylation/methylation with progression or regression events.
However, in order to increase the visual interpretation of the increased methylation levels in IM (data summarized in Table 1), we have added Figure S4 containing average methylation values for ZNF793 and RPRM in NM, precursor lesions and intestinal type of gastric cancer. It should be noticed that samples are corresponded to the discovery cohort (arrays) and not from the observational study.
New Supplementary Figure: “Figure S4. Methylation level of ZNF793 and RPRM genes in normal mucosa, precursor lesions and intestinal type of gastric cancer. Methylation data are corresponded to β-values from array (cg02711801 for ZNF793 and cg00341742 for RPRM). Horizontal bars represent the median of each sample group (p<0.0001 determined by a two-tailed Wilcoxon-Mann- Whitney test). CAG, multifocal chronic atrophic gastritis; IM, intestinal metaplasia; NAG, non-atrophic gastritis; NM, normal mucosa.”
Comment R1.8. The authors claim that “the proportion of methylation described in IM samples was higher than in non-metaplastic samples both for ZNF793 (Fisher’s test, p-value=0.009194) and RPRM (Fisher’s test, p-value=0.04173) (Figure 4B)”, but these Data are not shown in Figure 4B.
Answer R1.8. We have added this data in Figure 4B in the new version of the manuscript.
Comment R1.9. What’s the method used for association analysis in Figure 4C, and patient numbers of progression and no progression in each group are not shown.
Answer R1.9. Figure 4C has been modified. We have included two new columns with the number of methylated cases associated with progression and no progression. Statistical analysis (Fisher’s test) is also indicated in the legend.
Comment R1.10. The authors identified 13 common hypermethylated CpGs between IM and intestinal type gastric cancer. Is there any association of other methylated sites with progression?
Answer R1.10. We agree with the reviewer that another interesting candidates could be explored. Our FFPE validation cohort (longitudinal study) was extremely value but limited in quantity. Consequently, we need to prioritize which genes would be validated. We selected RPRM and ZNF793 because they already have a well-known role in gastric cancer and they are epigenetically-regulated genes (by CpG methylation). Unfortunately, we are not able to analyse progression/regression events in the validation cohort.
We have clarified our selection criteria for the candidate genes ZNF793 and RPRM.
Page 9: “Taking in consideration this list we selected two candidate genes based on the following criteria: first, high methylation similarities in IM and intestinal type of gastric cancer; second, previously described roles in gastric tumorigenesis; and third, previous evidences of gene regulation by CpG methylation20,21. We selected three cg probes from Illumina methylation arrays upstream the TSS of ZNF793 (cg02717801, cg23296010, cg14732998) and three cg probes upstream the TSS of RPRM (cg00341742, cg15400238, cg00143045) (Figure 3A) to further validate the methylation of their regulatory regions”.

Reviewer 2 Report
In the manuscript, the authors evaluated aberrant methylation observed in non-tumor stage gastric mucosa and found two important candidate genes, ZNF793 and RPRM that predict gastric cancer development. Although, the sample number was small, the data must be useful for the future studies. The comments for this manuscript are as follows:
Major
- Please explain the location of samples of “precursor lesions (n = 264)”. It is not clear if one biopsy was taken from one patient. In Figure 4, why the number of ZNF793 gene (n = 258) and RPRM gene (n = 169) were different from 264 biopsies. Please explain and indicate the reasons of sample exclusion.
- The reason why ZNF793 and RPRM were selected was not clear though they found 13 candidate CpG sequences. Even if some of the genes are not related to tumorigenesis, they might be still useful for the prediction of the development of gastric cancer in the future.
- Figure 3A must show the data of background mucosa and the data of cancer lesions with sample size about 12. What samples were used for these analyses?
- Figure 3B showed that RPRM and ZNF793 methylation levels were even high in antral intestinal metaplasia compared to intestinal metaplasia of body and cardia. Therefore, the gastric location of biopsy should be considered to assess the levels of target DNA methylations.
Minor
Page 9, line 2. “Figure 1D” did not exist.
Page 9, line 6. “Inflammatory Bowel Disease” should be “inflammatory bowel disease”.
Author Response
Reviewer 2
In the manuscript, the authors evaluated aberrant methylation observed in non-tumor stage gastric mucosa and found two important candidate genes, ZNF793 and RPRM that predict gastric cancer development. Although, the sample number was small, the data must be useful for the future studies. The comments for this manuscript are as follows:
Major comments
Comment R2.1. Please explain the location of samples of “precursor lesions (n = 264)”. It is not clear if one biopsy was taken from one patient. In Figure 4, why the number of ZNF793 gene (n = 258) and RPRM gene (n = 169) were different from 264 biopsies. Please explain and indicate the reasons of sample exclusion.
Answer R2.1. Because it was described that CpG methylation depends on anatomic-region (e.g., Huang et al, Cancer Cell 2017 PMID 29290541) we have only used samples from antrum. We have clarified the location of the biopsies in methods as follows:
Page 3: “In brief, five specimens from the same patient were obtained according to the Sidney recommendations (one from the incisura angularis, two from the antrum, and two from the corpus of the stomach). Diagnoses were made with one slide stained with hematoxylin and eosin and biopsy fragments were classified according to the most advanced lesion in the Correa cascade. Only the sample from antrum was selected for the methylation array study to avoid anatomic region-specific methylation”.
Although we have a total of 264 biopsies, we were not able to obtain PCR amplification of bisulfite- modified DNA for all samples. Technical limitations derived from formalin-fixed paraffin embedded tissue were observed and we were able to obtain 258 high-quality data for ZNF793 and 169 for RPRM. We have added this comment in results:
Page 12: “We analyzed the methylation status of RPRM and ZNF793 genes in the validation cohort of precursor lesions (n=264) obtained from an observational longitudinal study with a follow-up of 12 years to identify progression or regression events9 (Figure 4A). Because processing of formalin fixed-paraffin-embedded could be associated with DNA integrity problems, we were able to obtain high-quality Methylation- Specific PCR (MSP) results for 258 samples for ZNF793 and 169 samples for RPRM (Figure S4).”
Comment R2.2. The reason why ZNF793 and RPRM were selected was not clear though they found 13 candidate CpG sequences. Even if some of the genes are not related to tumorigenesis, they might be still useful for the prediction of the development of gastric cancer in the future.
Answer R2.2. We agree with the reviewer’s comment. As mentioned in the text, the 13 candidates could be interesting as they are associated with different types of cancer. As aforementioned, our FFPE cohort was extremely value but limited in quantity. Consequently, we need to prioritize which genes would be validated. We selected RPRM and ZNF793 because they already have a well-known role in gastric cancer. In addition, it was previously described that they are epigenetically-regulated genes (by CpG methylation). We have added this comment in the results section:
Page 9: “Taking in consideration this list we selected two candidate genes based on the following criteria: first, high methylation similarities in IM and intestinal type of gastric cancer; second, previously described roles in gastric tumorigenesis; and third, previous evidences of gene regulation by CpG methylation20,21. We selected three cg probes from Illumina methylation arrays upstream the TSS of ZNF793 (cg02717801, cg23296010, cg14732998) and three cg probes upstream the TSS of RPRM (cg00341742, cg15400238, cg00143045) (Figure 3A) to further validate the methylation of their regulatory regions”.
Comment R2.3. Figure 3A must show the data of background mucosa and the data of cancer lesions with sample size about 12. What samples were used for these analyses?
Answer R2.3. We apologyse for the mistake in the legend figure. There is not any data on diffuse type of cancer. Figure 3A includes 13 paired samples corresponding to intestinal subtype of gastric cancer and their normal adjacent tissues. We have lengthened the Y-Axis of the figure to increase the visibility of the paired samples. Methylation values were obtained as an average of the three cg probes selected from arrays. This information has been corrected in the Figure Legend:
“Figure 3. Methylation of ZNF793 and RPRM genes in intestinal type of gastric cancer and precursor lesions. (A) Methylation level of ZNF793 and RPRM genes in intestinal type of gastric cancer. Values from 13 adjacent-non-tumoral samples paired with the tumor sample are shown. Methylation values represent the average of β-value obtained for the three cg probes in the methylation array (paired t-test, two-tailored, p<0.05)”.
Comment R2.4. Figure 3B showed that RPRM and ZNF793 methylation levels were even high in antral intestinal metaplasia compared to intestinal metaplasia of body and cardia. Therefore, the gastric location of biopsy should be considered to assess the levels of target DNA methylations.
Answer R2.4. Yes, we agree with this comment. We apologize for not including this comment in the previous version. This question was also explained in comment R2.1. As it was described that the location of the biopsy influences CpG methylation of specific sequences, for the study we only used biopsies from antrum. We have added this comment in the text as follows:
Page 10: “Interestingly, hypermethylation of ZNF793 and RPRM genes in IM samples was not dependent on the anatomic region (antrum, cardia or body) as confirmed by analyzing an independent cohort of published dataset (GSE103186)[23] (Figure 3B). In accordance with previous observations, samples from antrum showed higher methylation level than samples from cardia or body [23]”.
Minor comments
Comment R2.5. Page 9, line 2. “Figure 1D” did not exist.
Answer R2.5. We apologize for the mistake. It corresponds to Figure 1B. It has been corrected in the text.
Comment R2.6. Page 9, line 6. “Inflammatory Bowel Disease” should be “inflammatory bowel disease”.
Answer R2.6. We have corrected the format in the text.

Reviewer 3 Report
Gomez et al analyzed the Methylation profiles within the precancerous gastric lesions as well as in gastric tumors.
The authors focused on a very interesting point. There are only minor points that should be clarified for the readers.
The authors analyzed at first phase intestinal and diffuse subtypes. In the introduction the authors stated that these subtypes are distinguishable at the histological level, however, the authors did not mention by whom they have confirmed these two sets, by pathologist? This should be stated.
Similarly, it is not clear if authors used adjacent mucosa in precursors lesions for methylation assessment. The same applies to primary tumors. Although this information can be found in workflow, in the text it is missing.
Did the authors analyze the cancer cell number within gastric tumors?
The statistical part is missing in the text. How authors also processed the expression data. Which housekeeping genes were used, etc.
Why authors used MSP for methylation assessment in ZNF793 and RPRM. According to Kristensen LS, Hansen (2009), MSP does not seem to be reliable enough for methylation studies and a subsequent verification of its results by supplementary techniques is inevitable.
The whole text should be checked by English native speaker several typos, incorrect words – circos in Figure 1; veerus in 3.2 section), or non-English style of writing numbers is used (for example in results, number of identified CpG sites, in Figure 3A p values).
The second sentence 3rd paragraph in the results section is not clear, is it about hyper or hypo DMCpG.
The authors mentioned that they further analyzed in the 3.3 result section their previous results on gastric cancer cell lines, however, the description of the methodology etc is missing.
In the legend to Figure 3A, the authors wrote that they present methylation levels of ZNF793 and RPRM genes in intestinal and diffuse type of gastric cancer, however, only intestinal results are presented.
My main question is, as the authors stated that 9 patients progressed to gastric cancer from precursor lesions analyzed in this study. Do the authors have access to those cancer samples to observe whether there is a different/the same methylation profile between precursors and gastric cancer within the same individual?
Minor:
Table 1 misses the abbreviation legend
Author Response
Reviewer 3
Gomez et al analyzed the Methylation profiles within the precancerous gastric lesions as well as in gastric tumors. The authors focused on a very interesting point. There are only minor points that should be clarified for the readers.
Comment R3.1. The authors analyzed at first phase intestinal and diffuse subtypes. In the introduction the authors stated that these subtypes are distinguishable at the histological level, however, the authors did not mention by whom they have confirmed these two sets, by pathologist? This should be stated.
Answer R3.1. Samples were provided for a Hospital Biobank Unit. Diagnosis were established by histologica studies performed by a pathologist at the Hospital Donostia. Briefly, from each block, slides were prepared and analysed using the haematoxylin-eosin, alcian blue (pH 2.5)-periodic acid Schiff (AB-PAS), and modified Giemsa stains. We have added this comment in Methods:
Page 3: “Primary tumors from gastric adenocarcinomas were obtained from The Basque Biobank for Research-OEHUN (Ref. CBVI239). Two samples pieces were extracted from each patient: one corresponds to the tumor and other from adjacent non-tumor gastric mucosae from the same patient. A pathologist performed the histologic diagnosis. For each block, slides were prepared and analysed using the haematoxylin-eosin, alcian blue (pH 2.5)-periodic acid Schiff (AB-PAS), and modified Giemsa stains”
Comment R3.2. Similarly, it is not clear if authors used adjacent mucosa in precursors lesions for methylation assessment. The same applies to primary tumors. Although this information can be found in workflow, in the text it is missing.
Answer R3.2. We have incorporated new sentences in the text to clarify the comparisons.
Page 5: “To identify the methylation differences among the normal mucosa and the intestinal or diffuse subtypes at specific sequences, we established a two-step data pro-cessing (Figure 1A). First, we filtered the samples by applying a threshold of > 0.33 change in average β-values and a false discovery rate (FDR) < 0.01 in an ANOVA test adjusted for multiple testing. As a result, we were able to establish a methylation sig-nature for the diffuse and intestinal subtypes compared to normal mucosa” […]
Page 7: “Consistent with previous reports[7,16], we also found that CpG methylation increas-es in IM compared to normal mucosa (down panel Figure 2B, Table S4).”
Comment R3.3. Did the authors analyze the cancer cell number within gastric tumors?
Answer R3.3. Unfortunately, we did not have this information for individual slides. Diagnosis is based on the Lauren classifications based on cellular type and differentiation. In spite that this is not the optimal situation, and that cellular heterogeneity exist, we do not expect different ratios of normal/tumoral content between intestinal or diffuse subtype, that is to say, the bias should be equally distributed.
In addition, we have included the necessity of dissection cellular heterogeneity in the discussion as follows:
Page 14: “We are aware that one of the main difficulties in epigenetic research is to discard tissue-specificity and the relative contribution of each cell type; an aspect especially relevant for biopsies taken in clinical environments. Single cell analysis to deconvolute sample heterogeneity should help to solve this question”.
Comment R3.4.The statistical part is missing in the text. How authors also processed the expression data. Which housekeeping genes were used, etc.
Answer R3.4. We apologize for the missing information. We have added the description of the expression analysis in Methods:
Page 3: “For quantitative RT-PCR assays, 2 μg of total RNA were converted to cDNA with the ThermoScriptTM RT-PCR System (Invitrogen) using Oligo-dT as primer. PCR amplifications were performed as follows: 0.20 μg of cDNA; 5 pmol of each primer and SYBRGreen PCR Master Mix (Applied Biosystems). Three measurements were analyzed using a Prism 7700 Sequence Detection (Applied Biosystems) instrument. All qRT-PCR were normalized using GAPDH expression as endogenous control. qRT-PCR primer sequences were: for ZNF793, 5’- CCA AGA GTG AGG CTG GTT TC -3’ (sense) and 5’- CCA AGG TCC TGT GGC TCT AA -3’ (antisense); for RPRM, 5’- CCC GCC AAG TTC CAA CAG -3’ (sense) and 5’- CTG TTG GCC AGG AAC AGG -3’ (antisense); for GAPDH, 5’-GAAGGTGAAGGTCGGAGTCA-3’ (sense) and 5’-TGGACTCCACGACGTACTCA -3’ (antisense).”
Comment R3.5. Why authors used MSP for methylation assessment in ZNF793 and RPRM. According to Kristensen LS, Hansen (2009), MSP does not seem to be reliable enough for methylation studies and a subsequent verification of its results by supplementary techniques is inevitable.
Answer R3.5. Thanks for the comment. We agree with the reviewer that many different assays have been developed with the goal of measuring DNA methylation in large cohorts and in the context of routine clinical diagnostics. Indeed, we have previously published (BLUEPRINT consortium, Nature Biotechnology 2016 PMID: 27347756) a study to compare the robustness of the different methylation assays and their use in clinical samples. The study, which includes 27 different methylation assays (absolute, relative or global methylation assays), demonstrated that qMSP was an appropriate assay to obtain a qualitative determination. As DNA methylation is a binary mark (i.e., a single CpG on a single allele in a single cell is either methylated or unmethylated), through its PCR amplification steps qMSP facilitates the determination of presence/absence of methylated sites on heterogeneous and/or partially degraded samples. Given the technical limitations of our FFPE cohort, we decide to apply qMSP to detect presence/absence of methylation. We agree that for quantification of methylation values (expressed as percentages or absolute numbers) other techniques are preferable. However, in our work we expressed methylation results as “methylated” or “unmethylated”, and consequently, we consider that qMSP is a valid technique.
Comment R3.6. The whole text should be checked by English native speaker several typos, incorrect words – circos in Figure 1; veerus in 3.2 section), or non-English style of writing numbers is used (for example in results, number of identified CpG sites, in Figure 3A p values).
Answer R3.6. Thanks for the comment. The new version has been checked for typos.
Comment R3.7. The second sentence 3rd paragraph in the results section is not clear, is it about hyper or hypo DMCpG.
Answer R3.7. We have re-write the sentence to facilitate its understanding as follows:
Page 5: “First, intestinal subtype showed a higher widespread hypomethylation of the genome. We identified 358 hypomethylated sequences in intestinal subtype compared to normal mucosa; while only 169 hypomethylated sequences were identified for diffuse subtype compared to normal mucosa.”
Comment R3.8. The authors mentioned that they further analyzed in the 3.3 result section their previous results on gastric cancer cell lines, however, the description of the methodology etc is missing.
Answer R3.8. Figure S3A contains methylation data obtained from methylation arrays (Illumina 450K) and published in reference 22 (Iorio F, et al. A Landscape of Pharmacogenomic Interactions in Cancer.Cell. 2016. PMID: 27397505). Methylation data were generated in our lab and are free available at Gene Expression Ommibus: GSE68379. For Figure S3A we used the three CpG probes selected from array as DMCpG. We have add this information in the text:
Page 9: “We also analyzed the methylation level of ZNF793 and RPRM genes in a set of gastric cancer cell lines (n=31) from which we had in house methylation data (free available at GEO:GSE68379[22]. Figure S3A showed methylation values using the three DMCpG probes for ZNF793 (cg02717801, cg23296010, cg14732998) and RPRM (cg00341742, cg15400238, cg00143045) obtained from Infinium 450K methylation arrays. Results confirmed that ZNF793 and RPRM CpG methylation is a frequent event in gastric cancer (Figure S3A)”
Comment R3.9 In the legend to Figure 3A, the authors wrote that they present methylation levels of ZNF793 and RPRM genes in intestinal and diffuse type of gastric cancer, however, only intestinal results are presented.
Answer R3.9. We apologize for the mistake. We have corrected the Figure legend:
“Figure 3. Methylation of ZNF793 and RPRM genes in intestinal type of gastric cancer and precursor lesions. (A) Methylation level of ZNF793 and RPRM genes in intestinal type of gastric cancer. Values from 13 adjacent-non-tumoral samples paired with the tumor sample are shown. Methylation values represent the average of β-value obtained for the three cg probes in the methylation array (paired t-test, two-tailored, p<0.05).”
Comment R3.10. My main question is, as the authors stated that 9 patients progressed to gastric cancer from precursor lesions analyzed in this study. Do the authors have access to those cancer samples to observe whether there is a different/the same methylation profile between precursors and gastric cancer within the same individual?
Answer R3.10. We agree with the reviewer that it would be very informative to have the methylation data of the tumor biopsy. Unfortunately, we do not have access to cancer tissue taken from patients enrolled in the study. Patients and biological samples were obtained from an observational longitudinal study of gastric precursor lesions (more information about the study could be found at PMID: 26630310).
Briefly, all patients with histological diagnoses of CAG, IM or dysplasia between 1995 and 2004 were identified from the Pathological Anatomy Department’s files of 10 public hospitals of Spain. For the end of follow-up gastroscopy of patients who accepted to participate in the study, five specimens were obtained by gastric endoscopy according to the Sydney recommendations to identify progression or regression in the precursor cascade. Patients who developed cancer were identified from medical history but the study does not include biopsy. Some limitations difficult this possibilities; first, different hospital services are involved (gastroenterology for precursor lesions but oncology for gastric cancer), and second, some patients have died during the period of the study (12 years).
Minor:
Comment R3.11. Table 1 misses the abbreviation legend
Answer R3.11. Abbreviations have been added to Table 1 legend.

Reviewer 4 Report
In this study, the authors showed that (1) the CpG methylation signatures of intestinal and diffuse types of gastric cancer are different and (2) H. pylori infection influenced DNA methylation in early precursor lesions but not in intestinal metaplasia. (3) CpG methylation at ZNF793 and RPRM promoters was assessed in a large cohort of patients (n=264) with a 12-year follow-up to identify the prognostic significance of these two methylated genes in predicting progression.
My major concern is, in contrast to the statements "We identified CpG methylation at ZNF793 and RPRM promoters as potential biomarker for the identification of patients at risk of cancer progression from intestinal metaplasia" and "we provide a proof-of-concept to investigate the practical utilities of the quantification of DNA methylation at candidate genes as a marker for disease progression", the authors' own data does not support any association between CpG methylation at ZNF793 and RPRM promoters with progression to gastric cancer from IM. Therefore, the current report has low translational value.
Author Response
Reviewer 4
In this study, the authors showed that (1) the CpG methylation signatures of intestinal and diffuse types of gastric cancer are different and (2) H. pylori infection influenced DNA methylation in early precursor lesions but not in intestinal metaplasia. (3) CpG methylation at ZNF793 and RPRM promoters was assessed in a large cohort of patients (n=264) with a 12-year follow-up to identify the prognostic significance of these two methylated genes in predicting progression.
Comment R4.1. My major concern is, in contrast to the statements "We identified CpG methylation at ZNF793 and RPRM promoters as potential biomarker for the identification of patients at risk of cancer progression from intestinal metaplasia" and "we provide a proof-of-concept to investigate the practical utilities of the quantification of DNA methylation at candidate genes as a marker for disease progression", the authors' own data does not support any association between CpG methylation at ZNF793 and RPRM promoters with progression to gastric cancer from IM. Therefore, the current report has low translational value.
Answer R4.1. We thank the reviewer for the comment. We agree that we do not have enough cases of progression to gastric cancer as to conclude that ZNF793 and RPRM is a biomarker for progression. We have town down the message along the text, especially in the abstract and conclusions. For examples:
- In simple summary. We have introduced “We identified CpG methylation at ZNF793 and RPRM promoters as a common event in intestinal metaplasia and intestinal type of gastric cancer”…..instead of “We identified CpG methylation at ZNF793 and RPRM promoters as potential biomarker for the identification of patients at risk of cancer progression from intestinal metaplasia”
- In conclusion and abstract. We have introduced “Herein, we provided molecular clues to further investigate the practical utilities of the quantification of REPRIMO and ZNF793 DNA methylation level as a marker for gastric cancer risk. Future longitudinal cohort studies will be especially valuable to validate their involvement in gastric cancer progression“ ….. instead of “we provided a proof-of-concept to investigate the practical utilities of the quantification of REPRIMO and ZNF793 DNA methylation level as a marker for gastric cancer risk”.
We are aware that future studies with large cohorts are needed to increase the number of cases that progress to cancer. In our study, we obtained 9 cases that progress to cancer, which represents an incidence rate of 2.5 per 1000 person/year. This rate is in accordance with previous estimations in European populations (1.75 cases per 1000 person year).
We would like to highlight the difficulties to obtain non-tumoral cohorts with long follow-up periods because this follow-up is not included in clinical routine. In our multicentric study, first we identified all patients with a preliminary histological diagnoses of precursor lesions between 1995 and 2004 from the Pathology Department’s files at 9 Spanish Hospitals. After revision of inclusion criteria, selected patients were invited for a second biopsy. Taking into consideration that this second biopsy was not included into clinical routine and that many patients have a non-tumoral stage at the moment of the study, the participation rate (64%) of the study was considered a success.
As we explained in discussion, we are proposing a candidate to be further explored in future larger studies. Although the translational value is pending of further validation, the potential of RPRM and ZNF793 is supported by their well-known role in gastric cancer. Interestingly, they are also epigenetically regulated genes. Together with our evidences, we strongly believe that they are interesting candidates. It should be noted that detection of the CpG methylation level of a specific region could be easily performed by simple approaches without complex handling requirements and into an acceptable cost/effectiveness margin which facilitates the development of larger studies in clinical environments.

Round 2
Reviewer 1 Report
Major comments
- Student’s t test should be used to compare two groups of small sample size consecutive variable which fit normal distribution.
- Figure 2b, the groups should be presented corresponding to the sequence NM, NAG, CAG, IM reflecting increasing severity of precursor lesions. Also, the GC group should also be added. A non-parametric test should be used (such as Wilcoxon). Additionally, I do not agree to the authors interpretation that “this result reflects the inexistence of a genome-wide reorganization of the methylome initiated at early stages of precursor lesions”. To my understanding, the beta values only reflect the mean levels of methylation in a given cohort. Thus, methylation patterns can be completely different and reorganized, but could still present similar overall beta values.
- In Table S4, why are there p values such as 1.21E+08, 1.54E+09 and so on? Shouldn’t it be 1.21E-08? The same potential errors in Table S3. The authors should rank the target ID by ascending p value and all the p values should be shown with three the same format (e.g. 0.007, where three numbers behind the 0, are sufficient) or use scientific notation. All the tables should use a standard three-line table as shown in S3. Why do other tables have a different format? All tables lack of footnotes. I apologize for being blunt but this gives the manuscript a very sloppy appearance.
- Figure 2E, the authors corrected the horizontal bar, but according to the heatmap features, infected samples should be clustered together (the two unknown samples should be clustered together with uninfected samples, and the infected sample with the other infected samples). Why didn’t the authors adjust the samples’ order according to the clustering?
Also here, I do not agree with the authors conclusion (page 15 first paragraph: “These clinical observations are in agreement with our observation that H. pylori presence or absence in high-risk precursor lesions does not change DNA methylation. In contrast, earlier lesions showed differential methylation between H. pylori positive or negative patients. Although larger follow-up studies are needed, these evidences opens future studies to address whether eradication of H. pylori by therapeutic strategies before CAG should prevent epigenome reprogramming towards a cancer signature.”
This is in my eyes a misinterpretation of the data: The fact that methylation patterns differ between infected and non-infected in CAG, but do not differ in IM, simply reflects that the methylation changes observed in CAG are dependent on or influenced by H. pylori infection. In contrast, IM is (as the name implies) a metaplasia, where cells if intestinal differentiation occur in the stomach. This is due to a profound trans-differentiation process, which obviously goes along with changes in methylation. This change in differentiation - according to the findings shown here – is independent of H. pylori infection. However, clinical experience informs us that this has NO consequence for treatment, since in both cases, eradication of H. pylori will in most cases prevent cancer development.
- I didn’t see any reversion in Figure S3B. And Figure S3A doesn’t fit its descriptions in the revised manuscript. And there is no Figure S5 in the supplementary file. Maybe the authors provided a wrong supplementary file.
- I don’t see any revision in Table 1. NM should be used as unified reference for all groups.
- In Figure 4C, the results showed no differences of methylation distribution in progression and non-progression groups. But this doesn’t mean there are no associations between methylation and progression. The authors should try multivariable logistic regression with progression as dependent variable and methylation as independent variable and adjust for potential confounders. Only then they can conclude on association.
- At several places in the manuscript, the authors claim the clinical significance of their work. Since no therapeutic strategy is presented or can ecen be thought of to interfere selectively with the methylation alterations observed, I think this manuscript is very far from clinical translation and rather presents basic research. Such overstatements should be removed.
- In summary, this is a very sloppy revision, and I don’t have the feeling that the authors have taken the comments serious. In this shape, the manuscript should not be published.
Author Response
Major comments
Comment R1.1. Student’s t test should be used to compare two groups of small sample size consecutive variable which fit normal distribution.
Response R1.1. We have introduced the results for Student’s t test.
Page 6: The distribution of clinical variables in the two groups was analyzed, demonstrating that there was no bias on the two arms of comparison for age (Student’s t- test, p=0.2879), for tumor location (Fisher's exact test, p=0.1789) and for metastasis (Fisher's exact test, p-value= 0.673).
Page 8: The distribution of age and gender in the comparison groups was analyzed, demonstrating that there were no significant differences among precursor lesions in the following comparisons: NM vs NAG (Student’s t- test, p=0.78 for age; Fisher's exact test, p=0.3007 for gender); NM vs CAG (Student’s t- test, p=0.31 for age; Fisher's exact test, p=1 for gender) and NM vs IM (Student’s t- test, p=0.26 for age; Fisher's exact test, p=0.4648 for gender).
Comment R1.2. Figure 2b, the groups should be presented corresponding to the sequence NM, NAG, CAG, IM reflecting increasing severity of precursor lesions. Also, the GC group should also be added. A non-parametric test should be used (such as Wilcoxon). Additionally, I do not agree to the authors interpretation that “this result reflects the inexistence of a genome-wide reorganization of the methylome initiated at early stages of precursor lesions”. To my understanding, the beta values only reflect the mean levels of methylation in a given cohort. Thus, methylation patterns can be completely different and reorganized, but could still present similar overall beta values.
Response R1.2. Figure 2B has been reordered to show the sequential order of precursor lesions: NM, NAG, CAG, IM, GC. In addition, following reviewer’s suggestions, we have suppressed the second part of the sentence in page 8. Only the part referring to “Unsupervised overall beta values of the precursor lesions do not show significant differences (Figure 2B)” has been maintained in the text.
Comment R1.3. In Table S4, why are there p values such as 1.21E+08, 1.54E+09 and so on? Shouldn’t it be 1.21E-08? The same potential errors in Table S3. The authors should rank the target ID by ascending p value and all the p values should be shown with three the same format (e.g. 0.007, where three numbers behind the 0, are sufficient) or use scientific notation. All the tables should use a standard three-line table as shown in S3. Why do other tables have a different format? All tables lack of footnotes. I apologize for being blunt but this gives the manuscript a very sloppy appearance.
Response R.1.3. Accordingly to reviewer’s suggestion, we have: (1) reordered the list using ascending p-values; (2) used a three decimals system for p-values; (3) standardized the format in Table S3, S4 and S5.
Comment R1.4. Figure 2E, the authors corrected the horizontal bar, but according to the heatmap features, infected samples should be clustered together (the two unknown samples should be clustered together with uninfected samples, and the infected sample with the other infected samples). Why didn’t the authors adjust the samples’ order according to the clustering?
Answer R1.4. The cluster is performing by ordering columns and raws, so based also in raw data this combination of columns is possible. It should be highlighted that clustering is a visual representation and that multitesting analysis segregates two groups. A more detailed explanation about clustering processing is provided bellow: Clustering in R is an unsupervised learning technique in which the data set is partitioned into several groups called as clusters based on their similarity. All the objects in a cluster share common characteristics. We used the M3C package to correct for the internal bias of consensus clustering by using a Monte Carlo simulation, M3C can also test the null hypothesis K=1. The logic behind the Monte-Carlo consensus clustering algorithm is that in the face of resampling the ideal clusters should be stable, thus any pair of samples should either always or never cluster together. We can use this principle to infer the optimal number of clusters (K). This works by examining cluster stability from K=2 to K=10 during resampling. To do this for every K, we calculate the consensus rates for all sample pairs, which is the fraction of times a pair of samples cluster together. This gives us a consensus matrix for every K, which is symmetrical and in the range 0 to 1. A matrix entirely of 1s and 0s will represent perfect stability for a given K. We can simply compare stability of these matrices from K=2 to K=10 to determine the optimal K. Hence, the cluster depicted is completely assessed for pval<0.05 rejecting the null hypothesis of K=1, which means that clusters are completely stable. We cluster using columns and rows, which means if we order artificially the clusters by columns as suggested, hence we will lose all the row dendrogram depicted.
Comment R1.5. Also here, I do not agree with the authors conclusion (page 15 first paragraph: “These clinical observations are in agreement with our observation that H. pylori presence or absence in high-risk precursor lesions does not change DNA methylation. In contrast, earlier lesions showed differential methylation between H. pylori positive or negative patients. Although larger follow-up studies are needed, these evidences opens future studies to address whether eradication of H. pylori by therapeutic strategies before CAG should prevent epigenome reprogramming towards a cancer signature.” This is in my eyes a misinterpretation of the data: The fact that methylation patterns differ between infected and non-infected in CAG, but do not differ in IM, simply reflects that the methylation changes observed in CAG are dependent on or influenced by H. pylori infection. In contrast, IM is (as the name implies) a metaplasia, where cells if intestinal differentiation occur in the stomach. This is due to a profound trans-differentiation process, which obviously goes along with changes in methylation. This change in differentiation - according to the findings shown here – is independent of H. pylori infection. However, clinical experience informs us that this has NO consequence for treatment, since in both cases, eradication of H. pylori will in most cases prevent cancer development.
Answer R1.5. We thanks the reviewer for the comment. We agree with the reviewer that CpG methylation in IM samples could be associated with additional factors such as the change of cellular phenotype. Indeed, this is the rationales of the study: the identification of CpG methylation differences in IM that resemble the cancer phenotype. The influence of Hp in IM samples could be “unmasked” by the regulation of cellular differentiation. However, the main message of the paper in relation to Hp infection is focused on the importance of considering epigenetic reprogramming in earlier lesions (CAG) and its connection with Hp. We emphasized the need of considering earlier precursor lesions in prevention strategies as the epigenetic reprogramming should be initiated at this step. Of course, this does not exclude prevention strategies in IM. Accordingly to reviewer comment, we have clarified this information in discussion:
Page 15: “Our study showed that the alteration of CpG methylation is induced in CAG and maintained during IM. Our observation is in accordance with previous results where no detectable differences in global DNA methylation between high-risk gastric cancer patients with and without active H. pylori infection were detected suggesting that methylation changes are irreversible once it has taken place [36]. It should be noted that another CpG methylation changes associated with cellular trans-differentiation in intestinal metaplasia could occur independently of H. pylori.”
Page 16: “They found that the incidence of gastric cancer development at the high-risk population after a 7.5 years follow-up was similar between participants receiving H. pylori eradication treatment and those receiving placebo. However, in the subgroup of H. pylori patients without precancerous lesions, eradication of H.pylori significantly decreased the development of gastric cancer [37]. These clinical observations are in agreement with our observation that H. pylori presence or absence in earlier lesions (eg, CAG) influences CpG methylation. Although larger follow-up studies are needed, these evidences open future studies to address whether eradication of H. pylori by therapeutic strategies before CAG should prevent epigenome reprogramming towards a cancer signature. These strategies could complement current interventions aimed to eradicate H. pylori in IM.”
Comment R1.6. I didn’t see any reversion in Figure S3B. And Figure S3A doesn’t fit its descriptions in the revised manuscript. And there is no Figure S5 in the supplementary file. Maybe the authors provided a wrong supplementary file.
Answer R1.6. Sorry, but we do not understand the comment. In the second version of the manuscript, Figure S3 contains the information indicated in the text: S3A is corresponded to methylation in the panel of cancer cell lines and Figure S3B represent in vitro data after demethylating treatments. Figure S5 is also included and contains the MSP results.
Comment R1.7. I don’t see any revision in Table 1. NM should be used as unified reference for all groups.
Answer R1.7. In the new version, we have added the NM-pair in all cases (NM-NAG; NM-CAG, NM-IM, NM-GC). We also maintained the comparisons between two sequential precursor lesions for representing the natural progression of the cascade. Corrected Table 1 has been added.
Comment R1.8. In Figure 4C, the results showed no differences of methylation distribution in progression and non-progression groups. But this doesn’t mean there are no associations between methylation and progression. The authors should try multivariable logistic regression with progression as dependent variable and methylation as independent variable and adjust for potential confounders. Only then they can concluded on association.
Answer R1.8. Generalized Linear Model was conducted to perform a logistic regression in binary outcome (progression or non-progression) depending on sample methylation status, adjusting for sex and age. No significant p-values were found: For RPRM: All lesions pval= 0.993; IIM pval = 0.831; CIM pval=0.982 For ZNF793: All lesions pval= 0.873; IIM pval = 0.863; CIM pval=0.789
Comment R1.9. At several places in the manuscript, the authors claim the clinical significance of their work. Since no therapeutic strategy is presented or can ecen be thought of to interfere selectively with the methylation alterations observed, I think this manuscript is very far from clinical translation and rather presents basic research. Such overstatements should be removed.
Answer R.1.9. We agree with the reviewer that further studies need to be performed to validate our results in clinical settings. As most of the discovery studies, the manuscript starts with basic research (description of the –omics associated to the epigenome) and ends by proposing two candidate genes to be further investigated. The message is to provide new interesting candidates to be validated in larger clinical settings. Accordingly, we have introduced in the text sentences as:
- In sum, our result supports the need to investigate the practical utilities of the quantification of DNA methylation at candidate genes as a marker for disease progression […]
- - Our results, by demonstrating epigenetic-associated silencing of cancer-related genes (ZNF793 and RPRM), provide a molecular mechanism to explain the increased risk of IM to progress to intestinal type of gastric cancer. Extension of these findings is needed and future studies in larger populations to increase the number of cases with cancer progression are required.
- - Although larger follow-up studies are needed, these evidences open future studies to address whether eradication of H. pylori […]

Reviewer 2 Report
I am satisfied with the corrections.
Author Response
We thank the reviewer for the positive comments.
Reviewer 4 Report
The authors have now addressed my previous concerns.
Author Response

(The authors gave the same response as above.)
